# SAMPLE-EFFICIENT QUALITY-DIVERSITY BY COOPERATIVE COEVOLUTION

**Ke Xue**[1,2*], **Ren-Jian Wang**[1,2*], **Pengyi Li**[3], **Dong Li**[4], **Jianye Hao**[3,4], **Chao Qian**[1,2†]

[1] National Key Laboratory for Novel Software Technology, Nanjing University, China
[2] School of Artificial Intelligence, Nanjing University, China
[3] School of Computing and Intelligence, Tianjin University, China
[4] Huawei Noah's Ark Lab, China

## ABSTRACT

Quality-Diversity (QD) algorithms, as a subset of evolutionary algorithms, have emerged as a powerful optimization paradigm with the aim of generating a set of high-quality and diverse solutions. Although QD has demonstrated competitive performance in reinforcement learning, its low sample efficiency remains a significant impediment for real-world applications. Recent research has primarily focused on augmenting sample efficiency by refining the selection and variation operators of QD. However, one of the less considered yet crucial factors is the inherently large-scale issue of the QD optimization problem. In this paper, we propose a novel Cooperative Coevolution QD (CCQD) framework, which decomposes a policy network naturally into two types of layers, corresponding to representation and decision respectively, and thus simplifies the problem significantly. The resulting two (representation and decision) sub-populations are coevolved cooperatively. CCQD can be implemented with different selection and variation operators. Experiments on several popular tasks within the QDax suite demonstrate that an instantiation of CCQD achieves approximately a 200% improvement in sample efficiency. Our code is available at `https://github.com/lamda-bbo/CCQD`.

## 1 INTRODUCTION

Generating a diverse and high-quality set of solutions is of paramount importance across a wide range of tasks, including Reinforcement Learning (RL) (Conti et al., 2018; Eysenbach et al., 2018; Parker-Holder et al., 2020), combinatorial optimization (Do et al., 2022; Nikfarjam et al., 2022), robotics (Cully et al., 2015; Salehi et al., 2022), and human-AI coordination (Lupu et al., 2021; Cui et al., 2021). Quality-Diversity (QD) algorithms (Cully & Demiris, 2018; Chatzilygeroudis et al., 2021), which are a subset of Evolutionary Algorithms (EAs) (Bäck, 1996), have emerged as a potent optimization paradigm for this challenging task. Specifically, a QD algorithm maintains a solution set (i.e., archive), and iteratively performs the following procedure: selecting a subset of parent solutions from the archive, applying variation operators (e.g., crossover and mutation) to produce offspring solutions, and finally using these offspring solutions to update the archive. The impressive performance of QD algorithms has been showcased in various RL tasks, such as exploration (Ecoffet et al., 2021; Miao et al., 2022), robust training (Kumar et al., 2020; Tylkin et al., 2021; Yuan et al., 2023b), environment generation (Fontaine et al., 2021; Bhatt et al., 2022; Zhang et al., 2023), open-ended learning (Standish, 2003; Soros & Stanley, 2014; Yuan et al., 2023a) etc.

Though able to achieve a diverse set of high-quality solutions, the sample efficiency of QD algorithms is not satisfactory (Chalumeau et al., 2023a). This issue can be alleviated by their high parallelization capabilities (Lim et al., 2023a), but remains a significant drawback, especially for applications with expensive evaluation. For example, each evaluation entails a substantial computational cost in environment generation (Dennis et al., 2020; Parker-Holder et al., 2022), making that the QD algorithm must resort to techniques such as surrogate models (Bhatt et al., 2022; Lim et al., 2022; Bhatt et al., 2023) to curtail computational costs. Recent research has tried to improve sample

---

*Equal contribution
†Correspondence to Chao Qian <qianc@nju.edu.cn>

efficiency from the algorithmic perspective of QD, such as refining selection (Cully & Demiris, 2018; Sfikas et al., 2021; Wang et al., 2022; 2023a) and variation (Nilsson & Cully, 2021; Tjanaka et al., 2022; Pierrot et al., 2022a; Faldor et al., 2023; Batra et al., 2024) operators.

The optimization problem of QD is indeed challenging. Typically, QD requires obtaining many solutions (e.g., 1024), and it is desirable for these solutions to be of high quality and diverse. What is even more challenging is that the dimensionality of solutions is usually very high. For example, when QD is applied to RL tasks (i.e., QD-RL, which is what we considered in this paper), a solution (e.g., a three-layer policy network) can involve tens of thousands of parameters (Chalumeau et al., 2023a). In this paper, we emphasize that one reason for the low sample efficiency of QD algorithm is the excessively large optimization space: optimizing complex single solution while simultaneously maintaining thousands of such solutions. If we can simplify the optimization problem of QD, its sample efficiency will be improved.

Fortunately, we observe that the solutions of a QD problem do not need to be completely different to achieve diversity. In QD-RL, if we can decompose the policy network and share some parts, then some common and useful knowledge can be shared, thus the optimization problem of QD will be greatly simplified. However, it is non-trivial to achieve this. Firstly, *How to decompose the policy network* is an important question. Besides, sharing too many parts (i.e., too many solutions share too many layers) can hinder diversity, which deviates from the goal of QD. Thus, another critical question is *How to balance the problem simplification and diversity maintenance?*

To address the issues raised above, we propose a novel framework called Cooperative Coevolution QD (CCQD), which is based on Cooperative Coevolution (Potter & Jong, 2000) and follows the divide-and-conquer approach. To the best of our knowledge, though many CCEAs have been proposed before (Ma et al., 2019), how to apply CC to simplify the optimization problem of QD has not been fully explored, particularly the design of decomposition strategy, which plays a critical role in CC (Omidvar et al., 2014a;b). Thanks to recent research, which has demonstrated the distinct functions of different layers of neural networks (Chung et al., 2019; Dabney et al., 2021; Zhou, 2021; Hao et al., 2023; Li et al., 2023): the front layers are used for state representation, while the following layers are used for decision-making. For example, ERL-Re$^2$ (Hao et al., 2023) achieves efficient knowledge sharing within the population by decomposing the policy network into shared state representations and independent linear representations. Building on this observation, CCQD decomposes each policy network in the population into two types of layers, a representation part and a decision part, resulting in two corresponding sub-populations, which are then coevolved cooperatively. Compared with the decision part, which is required to have diverse behaviors in the context of QD, the representation part contains a significant amount of common and shareable knowledge (Hao et al., 2023). Thus, we employ a representation population size that is much smaller than the decision population size, e.g., 20 vs. 1024, in our experiments. This will yield two benefits: 1) it further simplifies the problem while still maintaining a certain degree of diversity, and 2) the representation population can naturally contain more critics (as opposed to a single critic[1] employed in many studies (Nilsson & Cully, 2021; Flageat et al., 2023a; Pierrot et al., 2022a; Tjanaka et al., 2022)), thus alleviating the bias in off-policy updating. Additionally, Policy-extended Value Function Approximator (PeVFA) (Tang et al., 2022) is used as the critic, which can provide better value function approximation.

Our proposed framework CCQD is general, which can be implemented with different parent selection, variation, and survivor selection operators. We provide an instantiation of CCQD using uniform parent selection, IsoLineDD and policy gradient variation, and vanilla archive-based survivor selection. We mainly conduct experiments on the popular QDax suite (Lim et al., 2023a; Chalumeau et al., 2023b), including unidirectional, omnidirectional, and maze-type environments. These tasks are commonly used in QD-RL research (Tjanaka et al., 2022; Chalumeau et al., 2023a) and provide a challenging benchmark for evaluating the performance of CCQD. The results demonstrate that CCQD outperforms the baseline method (Mouret & Clune, 2015) and several state-of-the-art methods (Nilsson & Cully, 2021; Tjanaka et al., 2022; Pierrot et al., 2022a; Pierrot & Flajolet, 2023) in terms of many important QD metrics (e.g., QD-Score). We also demonstrate the versatility of CCQD by the experiments on Atari Pong (Bellemare et al., 2013). The effectiveness of CCQD is further verified by ablation studies, hyper-parameter sensitivity analysis, and the illustration of the archive.

---

[1]In our paper, for the sake of convenience, we group all the critic networks of a policy gradient algorithm (such as the four critic networks in TD3) together and refer to them as a single critic.

## 2 BACKGROUND

### 2.1 QUALITY-DIVERSITY

QD algorithms (Cully & Demiris, 2018; Chatzilygeroudis et al., 2021) aim to discover a diverse set of high-quality solutions for a given problem. Let $\mathcal{X}$ represent the solution space, and $\mathcal{S} \subseteq \mathbb{R}^k$ denote the $k$-dimensional descriptor space. The objective of QD algorithms is to maximize a fitness (quality) function $f : \mathcal{X} \to \mathbb{R}$ while exploring the $k$-dimensional descriptor space $\mathcal{S}$ using a behavior descriptor function $\boldsymbol{m} : \mathcal{X} \to \mathcal{S}$. Taking the widely recognized QD algorithm, MAP-Elites (ME) (Cully et al., 2015; Mouret & Clune, 2015), as an example, it maintains an archive by discretizing the descriptor space $\mathcal{S}$ into $M$ cells $\{\mathcal{S}_i\}_{i=1}^M$ and storing at most one solution in each cell. The main steps of ME involve selecting parent solutions from the archive, generating offspring solutions through variation operators, evaluating the offspring solutions, and updating the archive (i.e., survivor selection). A detailed algorithmic description of ME can be found in Appendix A. ME aims to fill the cells with high-quality solutions, thereby formalizing the objective as maximizing the QD-Score, denoted as $\sum_{i=1}^M f(\boldsymbol{x}_i)$, where $\boldsymbol{x}_i$ represents the solution contained within the cell $\mathcal{S}_i$, i.e., $\boldsymbol{m}(\boldsymbol{x}_i) \in \mathcal{S}_i$. If a cell $\mathcal{S}_i$ does not contain a solution $\boldsymbol{x}_i$, then $f(\boldsymbol{x}_i)$ is considered as 0. For simplicity, the fitness value $f(\cdot)$ is assumed (or converted) to be non-negative to prevent the QD-Score from decreasing.

Recently, many works investigate the use of QD in complex scenarios, broadening the application scenario of QD algorithms. Uncertain QD (Grillotti et al., 2023; Flageat & Cully, 2023; Flageat et al., 2023b) takes the uncertainty of the evaluation process into account and improves the reproducibility of the archive. The Quality-Diversity Transformer (QDT) (Macé et al., 2023) compresses an entire archive into a single behavior-conditioning policy, which can be used for many downstream applications. Multi-Objective QD (MOQD) (Pierrot et al., 2022b) extends QD to solve the multi-objective optimization problems that need diversity. Quality-Similar Diversity (QSD) (Wu et al., 2023) considers generating a set of diverse policies at multiple quality levels, which can be used for curriculum learning (Narvekar et al., 2020) where the environment gradually increases curriculum levels from simple to complex. Recent studies (Wang et al., 2023b; Ding et al., 2023) learn the behaviors from human feedback, addressing the difficulty of diversity definition in many applications.

### 2.2 REINFORCEMENT LEARNING

Consider a Markov decision process (MDP), defined by a tuple $\langle \mathcal{S}, \mathcal{A}, \mathcal{P}, \mathcal{R}, \gamma, T \rangle$. At each time-step $t$, the agent uses a policy $\pi_{\boldsymbol{\theta}}$ to select an action $a_t \sim \pi_{\boldsymbol{\theta}}(s_t) \in \mathcal{A}$ according to the state $s_t \in \mathcal{S}$, the environment transits to the next state $s_{t+1}$ according to transition function $\mathcal{P}(s_t, a_t)$, and the agent receives a reward $r_t = \mathcal{R}(s_t, a_t)$. The return is defined as the discounted cumulative reward, denoted by $R_T = \sum_{t=0}^T \gamma^t r_t$ where $\gamma \in [0, 1)$ is the discount factor and $T$ is the maximum episode horizon. The goal of RL is to learn an optimal policy $\pi^*$ that maximizes the expected return. Meanwhile, in many complex RL tasks such as robust training (Kumar et al., 2020; Tylkin et al., 2021), the goal is more challenging, requiring to find a set of diverse policies with high expected returns. QD has been successfully applied to these tasks, where the solution $\boldsymbol{x}$ and fitness $f$ in QD correspond to the policy $\pi_{\boldsymbol{\theta}}$ and episode return $R_T$ in RL, respectively.

Among different policy optimization algorithms, Deep Deterministic Policy Gradient (DDPG) (Lillicrap et al., 2016) is a representative off-policy Actor-Critic algorithm, consisting of a policy $\pi_{\boldsymbol{\theta}}$ (i.e., the actor) and a state-action value function approximation $Q_{\boldsymbol{\psi}}$ (i.e., the critic), with the parameters $\boldsymbol{\theta}$ and $\boldsymbol{\psi}$ respectively. The critic is optimized with the temporal difference loss (Sutton & Barto, 2018), and the actor is updated by maximizing the estimated $Q$ value. The loss functions are defined as: $\mathcal{L}(\boldsymbol{\psi}) = \mathbb{E}_{\mathcal{D}}[(r + \gamma Q_{\boldsymbol{\psi}'}(s', \pi_{\boldsymbol{\theta}'}(s')) - Q_{\boldsymbol{\psi}}(s, a))^2]$ and $\mathcal{L}(\boldsymbol{\theta}) = -\mathbb{E}_{\mathcal{D}}[Q_{\boldsymbol{\psi}}(s, \pi_{\boldsymbol{\theta}}(s))]$, where the experiences $(s, a, r, s')$ are sampled from the replay buffer $\mathcal{D}$, $\boldsymbol{\psi}'$ and $\boldsymbol{\theta}'$ are the parameters of the target networks. Twin Delayed DDPG (TD3) (Fujimoto et al., 2018) improves DDPG by addressing overestimation issue mainly by clipped double-$Q$ learning, leading to significant improvement as variation operator in QD-RL (Nilsson & Cully, 2021; Flageat et al., 2023a; Wang et al., 2022; Lim et al., 2023b; Pierrot et al., 2022a; Pierrot & Flajolet, 2023; **?**).

## 2.3 SAMPLE-EFFICIENT QD

One drawback of QD is the low sample efficiency (Chalumeau et al., 2023a; Li et al., 2024). Here, we briefly introduce the recent works on improving the sample efficiency of QD, which can be mainly divided into two categories according to the improved components of QD (Cully & Demiris, 2018): how to select parent solutions from the archive (i.e., parent selection) (Cully & Demiris, 2018; Sfikas et al., 2021; Wang et al., 2022; 2023a), and how to update them (i.e., variation) (Colas et al., 2020; Fontaine et al., 2020; Nilsson & Cully, 2021; Tjanaka et al., 2022; Pierrot et al., 2022a; Flageat et al., 2023a). A detailed related work is provided in Appendix A.2 due to space limitation.

## 2.4 COOPERATIVE COEVOLUTION

Cooperative coevolution (CC) (Potter & Jong, 2000) has been shown suitable for large-scale optimization (Ma et al., 2019). It uses a divide-and-conquer approach to decompose a large-scale problem into several small-scale sub-components and evolves these sub-components cooperatively. Each sub-component corresponds to a sub-population in the coevolutionary process. The key factor of cooperative coevolution is the decomposition strategy, which decides how to decompose a solution of the problem to be solved, and will largely influence the performance. Cooperative coevolution has been successfully applied to multiple complex tasks, e.g., POET (Wang et al., 2019; 2020) coevolves the agents and the environments, MAZE (Xue et al., 2022) coevolves the agents and the partners in zero-shot human-AI coordination, CCEP (Shang et al., 2022), coevolves the filters for network pruning, and CCNCS (Yang et al., 2022) coevolves different agents by random grouping. The problem to be solved by QD is usually large-scale, e.g., the number of parameters of a three-layer policy network in RL can be tens of thousands (Chalumeau et al., 2023a; Tjanaka et al., 2022). In this work, we apply CC to simplify the optimization problem of QD for the first time, and employ a natural and efficient layer-based decomposition strategy. Different from CCNCS, CCQD considers the characteristics of QD and policy networks, using a layer-based decomposition strategy to divide the policy network into two parts with different functions, and maintains fewer representation parts.

## 3 CCQD METHOD

In this section, we introduce the proposed CCQD framework. We first give an overview of CCQD. Then, we introduce the decomposition strategy and the coevolution process of CCQD in Section 3.1 and Section 3.2, respectively. As shown in Figure 3, CCQD follows the main process of QD: parent selection, variation, evaluation, and survivor selection. The main difference between CCQD and QD algorithms is that CCQD decomposes each policy into two parts – a representation part and a decision part – and maintains two sub-populations that are cooperatively coevolved. The detailed algorithm process of CCQD is provided in Appendix A.

## 3.1 DECOMPOSITION

The main idea behind CCQD is to apply the divide-and-conquer technique of cooperative coevolution to QD. Although cooperative coevolution has achieved impressive success in large-scale optimization, it has been reported that it may lead to poor performance in non-separable problems if an appropriate decomposition strategy is not employed (Ma et al., 2019). CCQD utilizes the specific characteristics of the policy network for the non-trivial separation issue of cooperative coevolution. Considering the observation that different layers of the neural network have different functions (Chung et al., 2019; Dabney et al., 2021; Zhou, 2021; Hao et al., 2023), a natural decomposition strategy is to decompose the search space of a policy network by layer. In particular, a policy network is divided into two parts, where several front layers of the policy network are considered as the representation part, while the following layers are used as the decision part. For each part to be optimized, CCQD maintains one sub-population. Each solution in a sub-population represents the corresponding part of a policy network. The two sub-populations are cooperatively coevolved through parent selection, variation, evaluation, and survivor selection, which will be explained in detail in Section 3.2.

To achieve a diverse set of high-quality solutions, a large population size is usually required. However, considering that the representation part contains a significant amount of common and shareable information compared to the decision part (Zhou, 2021; Hao et al., 2023), which requires diverse behaviors, a large size for both sub-populations is not necessary. Thus, CCQD employs a representation

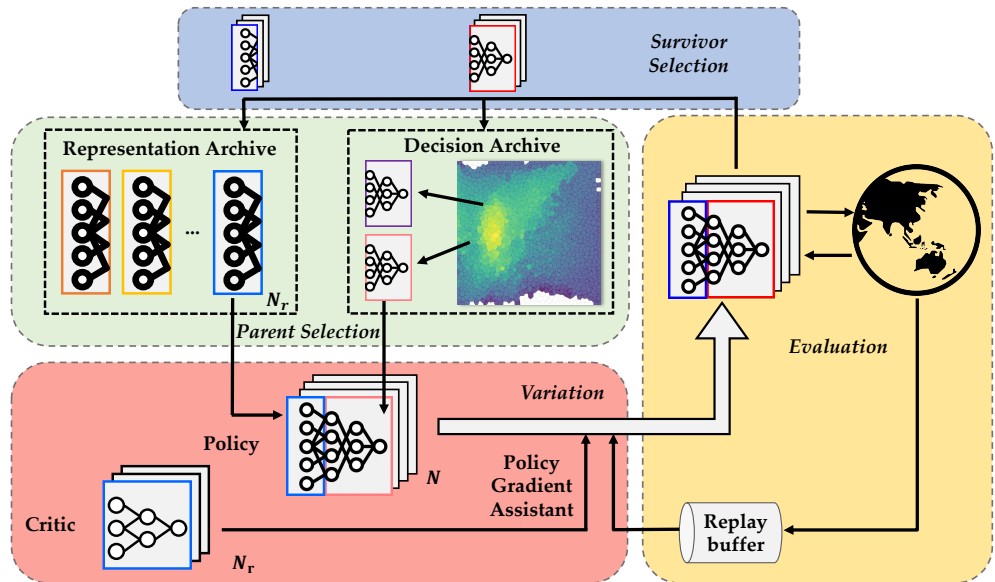

Figure 1: Illustration of the CCQD method, where CCQD decomposes a policy network into the representation and decision parts, and coevolves the two corresponding sub-populations through four sub-processes, i.e., parent selection, variation, evaluation, and survivor selection.

sub-population whose size is much smaller than that of decision sub-population, e.g., 20 vs. 1024 in our experiments. Using a small representation sub-population further simplifies the optimization problem of QD, which also allows CCQD to use more critics naturally and thus alleviates the bias in off-policy updating. This will be further explained in the Variation part of Section 3.2.

## 3.2 COEVOLUTION

The two sub-populations of CCQD have two different types of archives: a list-based archive for the representation part and a grid-based archive for the decision part. In the following, we will give a concrete implementation of CCQD by adopting specific parent selection, variation and survivor selection mechanisms, which is sufficient to achieve superior performance, as will be demonstrated in Section 4. But the CCQD framework itself is versatile, which can be combined with various existing QD techniques. Improving certain components of our implementation may lead to better performance, which will be studied in our future work.

**Parent Selection.** At each generation, we employ the uniform random selection strategy commonly used in QD, to choose a set of $N$ suitable parts (solutions) from the decision archive. Then, each selected decision part randomly selects a representation part from the representation archive to obtain a complete policy. Note that different decision parts may select the same representation part due to the reduced representation population size. This process ensures that the representation and decision parts are combined in a diverse manner.

**Variation.** We use the variation operator in PGA-ME (Nilsson & Cully, 2021; Flageat et al., 2023a) as our basic variation operator, i.e., one-half of the parent solutions is updated with the traditional evolutionary operator IsoLineDD, and the other half is updated with TD3. The TD3 operator focuses on optimizing quality, while the IsoLineDD operator is used to promote diversity. Note that there are two key differences in the usage of TD3 between CCQD and PGA-ME. 1) Each representation part in CCQD has its own critic, while PGA-ME only employs a single critic for all the policies in the population, leading to imprecise value function approximation and ultimately hindering sample efficiency (Hao et al., 2023). Thanks to the population size for the representation part being relatively small (e.g., 20 in our experiment), we can naturally maintain a critic for each representation part,

which allows us to alleviate the bias in off-policy updating and improve sample efficiency. 2) We use PeVFA (Tang et al., 2022) as our critic, which preserves the values of multiple policies. Concretely, given some representation embedding $\chi_\pi$ of policy $\pi$, a PeVFA parameterized by $\psi$ takes $\chi_\pi$ as input additionally, i.e., $Q_\psi(s, \chi_\pi, a)$. Through the explicit policy representation $\chi_\pi$, PeVFA provides better value generalization among policies, making it suitable for CCQD, where each representation part will be combined with multiple decision parts.

**Survivor Selection.** After generating and evaluating the offspring policies, CCQD updates the representation archive and decision archive by survivor selection. We use the survivor selection strategy of vanilla QD (Mouret & Clune, 2015) for the decision archive. That is, each offspring solution (decision part) is placed in its corresponding cell in the behavior space according to its behavior. If the cell is empty, the solution is kept directly; otherwise, the one with a higher quality between this solution and the solution occupying the cell is kept. Note that the complete policy network is saved in the decision archive, but only its decision part is selected by parent selection. When CCQD terminates, the output is directly the decision archive.

For the survivor selection of the representation archive, we use the simplest approach, which directly replaces a representation part that has been selected in parent selection by the updated one. After that, we check each part in the representation archive. If no solutions in the current decision archive are associated with a particular representation part, we replace that representation part with the one that is associated with the most solutions in the decision archive.

## 4 EXPERIMENT

To examine the performance of CCQD, we conduct experiments on the popular QDax suite[2] (Lim et al., 2023a; Chalumeau et al., 2023b), including unidirectional tasks (i.e., *Uni*), omnidirectional tasks (i.e., *Omni*), and *Maze*-type environments. The *Uni* tasks aim to generate a set of policies that move forward as fast as possible and are diverse in the frequency of the usage of each foot, where the reward is mainly determined by the forward speed of the robot, and the behavior descriptor is defined as the fraction of time each foot touches the ground. The *Omni* tasks aim to generate a set of policies that move to different directions while consuming as few energy as possible, where the reward is defined as the opposite of the energy consumption, and the descriptor is defined as the final position of the robot. The *Maze* tasks aim to train the point or robot to reach a specific target position in a maze, where the reward is defined as the opposite of the distance between the current position and the target position, and the behavior descriptor is defined as the final position of the agents. We mainly consider the following evaluation metrics: 1) **QD-Score**: The total sum of fitness values across all solutions in the archive. It reflects both the quality and diversity of the solutions, and is the most important metric to evaluate a QD algorithm (Pugh et al., 2016; Cully & Demiris, 2018); 2) **Coverage**: The total number of solutions in the archive. It can measure the exploration ability of a QD algorithm; 3) **Max Fitness**: The largest fitness value of solutions in the archive. It can measure the exploitation ability of a QD algorithm; 4) **QD-Score AUC** (Tjanaka et al., 2022): The area under the QD-Score curve. It measures the optimization efficiency of a QD algorithm.

To evaluate the effectiveness of CCQD, we compare it against the baseline method ME (Mouret & Clune, 2015), as well as four state-of-the-art approaches: PGA-ME (Nilsson & Cully, 2021; Flageat et al., 2023a), QD-PG (Pierrot et al., 2022a), OMG-MEGA (Tjanaka et al., 2022), and PBT-ME (Pierrot & Flajolet, 2023). Detailed explanations of these methods are provided in Appendix B.1. We use the CVT-ME archive (Vassiliades et al., 2018) to store solutions, which surpasses the vanilla grid-based ME approach on numerous tasks, particularly in the context of high-dimensional behavior space (Pierrot et al., 2022a; Pierrot & Flajolet, 2023). In our experiments, a policy is represented by a three-layer neural network, and CCQD uses the first layer as the representation part and the remaining two layers as the decision part. The size of representation archive is 20, and the decision archive is CVT-ME with size $1024$. The representation embedding $\chi_\pi$ of PeVFA used by CCQD relies on the parameter vector of the last layer of the decision part. All the methods are trained with a total budget of $1.5e8$ environment time-steps. The episode length is 250, which is consistent with previous works (Chalumeau et al., 2023a;b). The number of cells in the archive is 1024, and the number of generated offspring solutions in each generation is 100, except for PBT-ME, whose number is 320

---

[2]https://github.com/adaptive-intelligent-robotics/QDax

Table 1: QD-Score AUC ($\times 10^{12}$) of different methods on eight environments with episode length 250 and total timesteps $1.5e8$. The symbols '+', '$-$', and '$\approx$' indicate that the result is significantly superior to, inferior to, and almost equivalent to CCQD, respectively, according to the Wilcoxon rank-sum test with significance level 0.05. **Bold** and underline texts respectively denote the best and runner-up algorithms.

| Environment | ME | QD-PG | PGA-ME | OMG-MEGA | PBT-ME | CCQD |
|---|---|---|---|---|---|---|
| *Hopper Uni* | 84.17 $-$ | 75.20 $-$ | 93.25 $-$ | 91.47 $-$ | 81.32 $-$ | **96.75** |
| *Walker2D Uni* | 102.73 $-$ | 103.36 $-$ | 109.56 $-$ | 110.23 $-$ | 85.20 $-$ | **116.83** |
| *HalfCheetah Uni* | 343.79 $-$ | 323.44 $-$ | 388.24 $-$ | 392.61 $-$ | 425.16 $\approx$ | **432.83** |
| *Ant Uni* | 121.16 $-$ | 131.10 $-$ | 131.90 $-$ | 135.98 $\approx$ | 121.89 $-$ | **141.27** |
| *Humanoid Uni* | 119.36 $-$ | 125.09 $-$ | 116.36 $-$ | 117.43 $-$ | 97.61 $-$ | **132.51** |
| *Humanoid Omni* | 0.90 $-$ | 1.45 $-$ | 1.40 $-$ | 1.07 $-$ | 1.22 $-$ | **2.65** |
| *Point Maze* | 43.90 $-$ | 42.74 $-$ | 35.09 $-$ | 34.63 $-$ | 35.01 $-$ | **52.73** |
| *Ant Maze* | 105.90 $-$ | **164.94** $\approx$ | 141.64 $-$ | 146.46 $-$ | 132.47 $-$ | 157.03 |
| $+/-/\approx$ | 0/8/0 | 0/7/1 | 0/8/0 | 0/7/1 | 0/7/1 | / |
| Average Rank | 4.62 | 3.50 | 3.50 | 3.50 | 4.75 | **1.12** |

to be consistent with the original paper. For a fair comparison, all the methods use uniform parent selection. We report the median and the first and third quartile intervals across five identical seeds (1000, 2000, ..., 5000) for all algorithms on most tasks. For *Hopper Uni*, *Humanoid Omni*, and *Ant Maze*, we use ten seeds (1000, 2000, ..., 10000) because of the high randomness and difficulties in these environments. Detailed settings of experiments are provided in Appendix B.2.

**QD-Score AUC.** We first use the QD-Score AUC to compare the optimization efficiency of different methods, as shown in Table 1. By the Wilcoxon rank-sum test with significance level 0.05, CCQD is significantly better than almost any other methods on any of the eight environments, and has the best average rank. The three state-of-the-art methods, i.e., QD-PG, PGA-ME, and OMG-MEGA, outperform ME and PBT-ME (in average rank), and perform as the runner-up multiple times.

**Other metrics.** We then plot the QD-Scores, Coverage, and Max Fitness curves for different algorithms, allowing for a more comprehensive evaluation. As shown in Figure 2, CCQD has the best QD-Score across all environments except for *Ant Maze*. PBT-ME achieves the highest Max Fitness on *Ant Uni* and *Ant Maze*, but its poor Coverage results in a low QD-Score in these two environments. CCQD also exhibits advantages in both Max Fitness and Coverage in most environments, demonstrating its excellent performance. We also compare the average rank of Max Fitness. As shown in Table 9, CCQD achieves the highest average rank, and PBT-ME is the runner-up. In addition, we also compare the methods in the environments with episode length 1000, which is also a common setting in QDRL, as shown in Table 8, Table 10, and Figure 11. In this setting, CCQD still has the best average rank on QD-Score AUC and Max Fitness. We also consider archive profile and corrected metrics (Flageat et al., 2023b), as shown in Appendix C.2 and C.3, respectively.

**Ablation Studies.** We first examine the setting of our decomposition strategy, i.e., how many front layers are used as the representation part. We use a three-layer policy network, and thus have two options: 1 and 2, denoted as (1+2) and (2+1), respectively. Figure 3(a) shows that different settings can both significantly improve the sample efficiency, thereby validating the robustness of our decomposition strategy. However, for some complex problems, it may be necessary to allocate the proportion of representation and decision-making carefully. We also test CCQD with fixed paring in parent selection and that without checking in survivor selection of the representation archive. Note that our CCQD implementation uses random paring and also checks whether a representation part is associated with some decision part in the decision archive. Figure 3(b) shows the effectiveness of these employed strategies. Additional studies on number of representation population size, number of critics, and PeVFA are provided in Appendix C.4, C.5 and C.6, respectively.

**Archive Visualization.** An interesting question concerning CCQD is whether different representation parts truly distinguish different behaviors. To investigate this question, we plot the representation

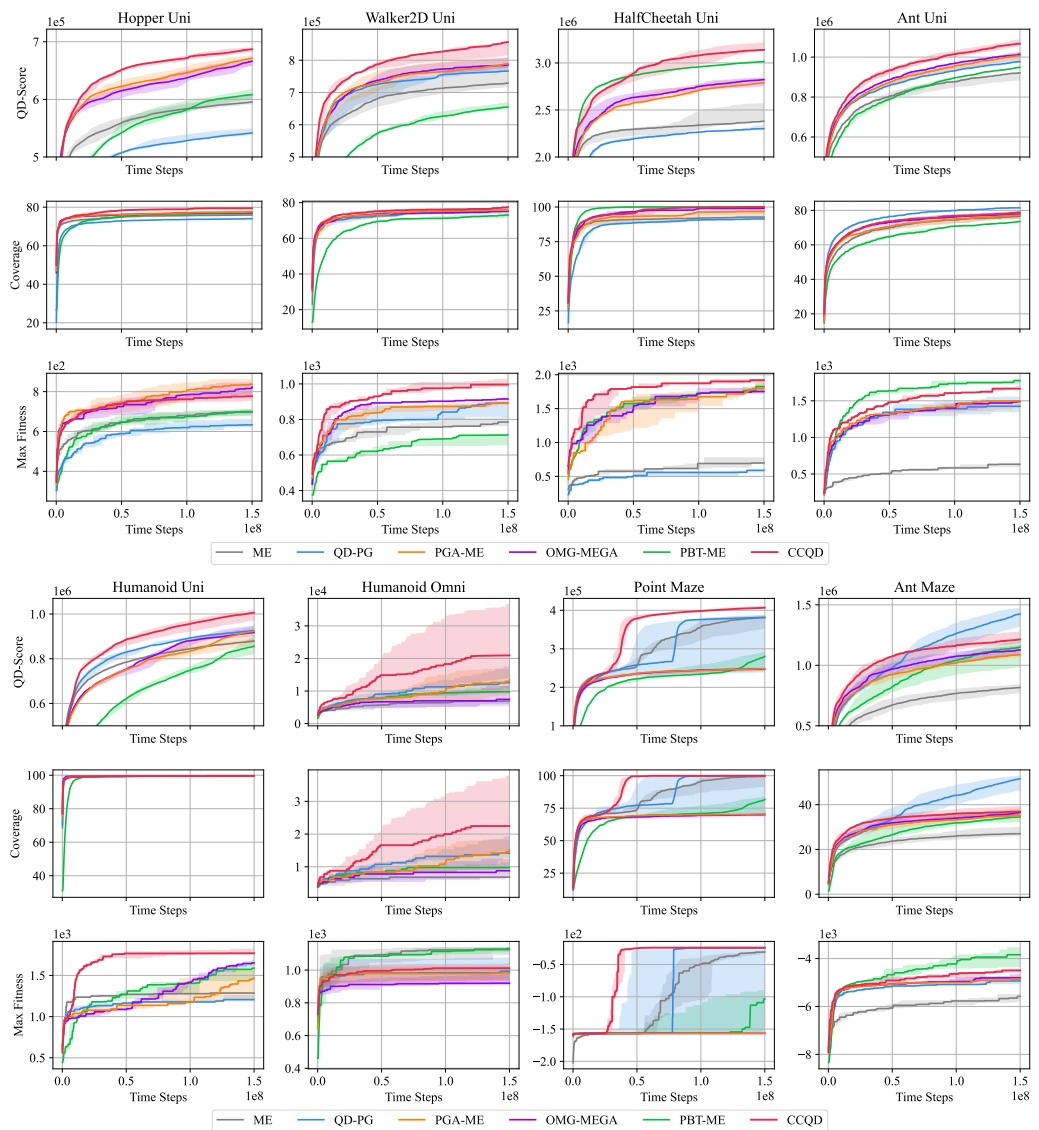

Figure 2: Performance comparison of CCQD with other methods in terms of QD-Score, Coverage, and Max Fitness on eight environments with episode length $250$ and total timesteps $1.5e8$. The medians and the first and third quartile intervals are depicted with curves and shaded areas, respectively.

part associated with each solution in the decision archive, where different representation parts are displayed using distinct colors. As illustrated in Figure 4 (a)-(b), solutions associated with the same representation part tend to have similar behavior descriptor values and are often located in nearby cells in the decision archive. Note that the colors in (a) represent the quality, while the colors in (b) represent the different representation parts. This finding suggests that different representation parts can recognize diverse state representations and discover various behaviors, which may explain the effectiveness of CCQD. Additional visualization results are provided in Appendix C.7.

**Experiments on Atari Pong.** To investigate the versatility of our framework, we conduct an experiment on a video game *Atari Pong* (Bellemare et al., 2013), which is a widely used benchmark. The state space is an image space whose dimension is about 100 times larger than that of QDax environments. We use DQN as variation operator and the corresponding variant of PGA-ME is DQN-ME. The policy network is changed to a DQN with three convolutional layers and two fully connected layers. We still use our layer-based decomposition strategies. We consider that the representation

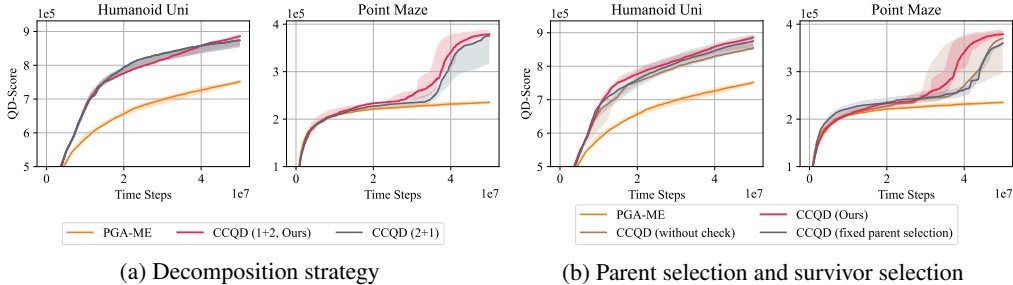

(a) Decomposition strategy    (b) Parent selection and survivor selection

Figure 3: (a) Analysis of decomposition strategy, where (1+2) denotes the representation part and decision part have 1 and 2 layers, respectively. (b) Ablation studies on parent and survivor selection.

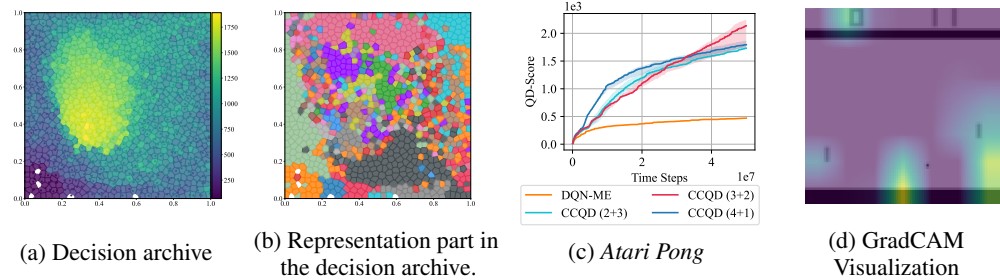

(a) Decision archive    (b) Representation part in the decision archive.    (c) *Atari Pong*    (d) GradCAM Visualization

Figure 4: (a) The decision archive in *Humanoid Uni*. (b) The representation parts associated with each decision part in the decision archive in *Humanoid Uni*. (c) Performance comparison on *Atari Pong*. (d) GradCAM analysis of the representation part.

part has 2, 3, and 4 layers, denoted as (2+3), (3+2), and (4+1), respectively. The detailed settings are provided in Appendix B.1. As depicted in Figure 4 (c), various configurations of CCQD consistently outperform DQN-ME, illustrating the versatility of the CCQD framework and its applicability to diverse tasks and network structures. Note that the number of parameters of *Atari Pong* (1.6 million) is much larger than that of Humanoid Uni (0.1 million). On this difficult task, our CCQD significantly outperforms QD. Furthermore, we use GradCAM (Selvaraju et al., 2017) to investigate *what does the representation part learn* in this video task, as shown in Figure 4 (d). We can find that the representation part focuses on the position of the ball and agents, which is helpful to finish the task.

## 5 CONCLUSION

This paper introduces the CCQD framework, which addresses the inherent large-scale challenge of QD by leveraging cooperative coevolution for the first time. Notably, CCQD is compatible with and can be combined with various existing QD techniques, such as parent selection, variation, and survivor selection. The simple implementation of CCQD presented in this paper has demonstrated impressive performance in multiple tasks, significantly improving sample efficiency. Exploring and employing better components could potentially lead to further improvements, and it would be an interesting avenue for future work. Additionally, the shared representation components in CCQD offer the potential for reducing the storage overhead of QD algorithms. In some computationally-constrained application tasks, combining the CCQD framework with archive distillation techniques (Macé et al., 2023) can reduce computational costs and achieve a sample and storage-efficient QD algorithm, which is an aspect we will investigate in future research. It would also interesting to investigate if the trained decision policies can be used as a robust sub-policy and be applied into other tasks, because they are trained on a variety of representations. One limitation of this paper is that we only demonstrate the effectiveness of CCQD through empirical studies, without delving into its properties from a theoretical perspective (Panait et al., 2008; Qian et al., 2024), which is also crucial.

ACKNOWLEDGEMENTS

We thank the anonymous reviewers for their insightful and valuable comments. We thank Li-He Li for helping drawing the workflow figure. This work was supported by the National Science and Technology Major Project (2022ZD0116600) and National Science Foundation of China (62276124).

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

# A    DETAILS OF QD ALGORITHMS

## A.1    QD

**ME.**    A detailed algorithm process of ME is provided in Algorithm 1. The main procedure of ME is to iteratively select parent solutions from the archive (i.e., line 6), generate offspring solutions by variation operators (i.e., line 7), evaluate the offspring solutions (i.e., line 9), and update the archive (i.e., survivor selection, line 10).

---

**Algorithm 1** MAP-Elites

---

**Input**: number $T$ of total generations, number $N$ of selected solutions in each generation
**Output**: archive $A$

1: Let $A \leftarrow \emptyset$, $t \leftarrow 1$;
2: **while** $t \leq T$ **do**
3:     **if** $t = 1$ **then**
4:         $\{\pi_i\}_{i=1}^N \leftarrow \texttt{Randomly\_Generatation}(N)$
5:     **else**
6:         $\{\pi_i\}_{i=1}^N \leftarrow \texttt{Parent\_Selection}(A, N)$;
7:         $\{\pi_i'\}_{i=1}^N \leftarrow \texttt{Variation}(\{\pi_i\}_{i=1}^N)$
8:     **end if**
9:     $\texttt{Evaluation}(\{\pi_i'\}_{i=1}^N)$;
10:    $A \leftarrow \texttt{Survivor\_Selection}(A, \{\pi_i'\}_{i=1}^N)$;
11:    $t \leftarrow t + 1$
12: **end while**
13: **return** $A$

---

**Algorithm 2** Survivor Selection of ME

---

**Input**: archive $A$, solutions $\{\pi_i'\}_{i=1}^N$
**Output**: updated archive $A$

1: **for** $i = 1 \rightarrow N$ **do**
2:     $j \leftarrow \texttt{Get\_Cell\_Index}(\boldsymbol{m}(\pi_i'))$;
3:     **if** $A_j$ is empty or $f(A_j) < f(\pi_i')$ **then**
4:         $A_j \leftarrow \pi_i'$
5:     **end if**
6: **end for**
7: **return** $A$

---

**CCQD.**    The proposed CCQD is a general framework that can be implemented with different components. In this paper, we provide an implementation using the components of PGA-ME, as shown in Algorithm 3. At the beginning, the decision archive $A$ and the replay buffer $B$ are created as empty sets in line 1, and $N$ initial solutions $\{\pi_i'\}_{i=1}^N$, along with the corresponding $N_r$ representation parts $\{\pi_i^{\mathrm{R}}\}_{i=1}^{N_r}$ and $N_r$ greedy actors $\{\pi_i^{\mathrm{G}}\}_{i=1}^{N_r}$, are randomly generated in line 4. After that, in each generation $t$ (where $t > 1$), CCQD first trains $N_r$ critics $\{Q_{\boldsymbol{\theta}}\}_{i=1}^{N_r}$ (each is parameterized with $\boldsymbol{\theta}$) with TD loss in line 6 and the greedy actors $\{\pi_i^{\mathrm{G}}\}_{i=1}^{N_r}$ with policy gradient in line 7. Note that during the process of training critics in line 6, $N$ decision parts $\{\pi_i^{\mathrm{D}}\}_{i=1}^N$ are randomly selected from the archive $A$ in each iteration to calculate the TD loss. Then, $N$ decision parts $\{\pi_i^{\mathrm{D}}\}_{i=1}^N$ are selected from the archive in line 8, and each of them is combined with a representation part in the representation archive $\{\pi_i^{\mathrm{R}}\}_{i=1}^{N_r}$, leading to $N$ complete policies in line 9. The two parts are updated with their corresponding variation operators in line 10 and line 11, respectively. After evaluating the generated offspring solutions $\{\pi_i'\}_{i=1}^N$ in line 13, we get the fitness and behavior of each solution. In line 14, the representation archive $\{\pi_i^{\mathrm{R}}\}_{i=1}^N$ and the decision archive $A$ are updated by survivor selection according to the fitness $f$ and the behavior descriptor vector $\boldsymbol{d}$ in line 15.

---

**Algorithm 3** CCQD

---

**Parameter**: number $T$ of total generations, representation population size $N_r$, number $N$ of generated solutions in each generation

**Output**: decision archive $A$ with the associated representation part

1:  Let $A \leftarrow \emptyset$, $B \leftarrow \emptyset$, $t \leftarrow 1$;

2:  **while** $t \leq T$ **do**

3:    **if** $t = 1$ **then**

4:      $\{\pi_i'\}_{i=1}^N, \{\pi_i^R\}_{i=1}^{N_r}, \{\pi_i^G\}_{i=1}^{N_r} \leftarrow \texttt{Randomly\_Generatation}(N, N_r)$

5:    **else**

6:      $\{Q_{\boldsymbol{\theta}}\}_{i=1}^{N_r} \leftarrow \texttt{Train\_Critics}(\{Q_{\boldsymbol{\theta}}\}_{i=1}^{N_r}, \{\pi_i^G\}_{i=1}^{N_r}, \{\pi_i^R\}_{i=1}^{N_r}, A)$;

7:      $\{\pi_i^G\}_{i=1}^{N_r} \leftarrow \texttt{Train\_Greedy\_Actors}(\{\pi_i^G\}_{i=1}^{N_r}, \{Q_{\boldsymbol{\theta}}\}_{i=1}^{N_r})$;

8:      $\{\pi_i^D\}_{i=1}^N \leftarrow \texttt{Select\_Decision\_Parts}(A, N)$;

9:      $\{\pi_i\}_{i=1}^N \leftarrow \texttt{Combine\_with\_Representation\_Parts}(\{\pi_i^R\}_{i=1}^{N_r}, \{\pi_i^D\}_{i=1}^N)$;

10:     $\{\pi_i'\}_{i=1}^N \leftarrow \texttt{Variation\_Decision\_Parts}(\{\pi_i\}_{i=1}^N, \{Q_{\boldsymbol{\theta}}\}_{i=1}^{N_r})$;

11:     $\{\pi_i'\}_{i=1}^N \leftarrow \texttt{Variation\_Representation\_Parts}(\{\pi_i'\}_{i=1}^N, \{Q_{\boldsymbol{\theta}}\}_{i=1}^{N_r})$

12:    **end if**

13:    $\texttt{Evaluation}(\{\pi_i'\}_{i=1}^N)$;

14:    $A, \{\pi_i^R\}_{i=1}^{N_r} \leftarrow \texttt{Survivor\_Selection}(A, \{\pi_i^R\}_{i=1}^{N_r}, \{\pi_i'\}_{i=1}^N)$;

15:    $t \leftarrow t + 1$

16:  **end while**

17:  **return** $A$ with the associated representation part

---

### A.2   SAMPLE-EFFICIENT QD

One drawback of QD is the low sample efficiency. Here, we briefly introduce the recent works on improving the sample efficiency of QD, which can be mainly divided into two categories according to the improved components of QD (Cully & Demiris, 2018): how to select parent solutions from the archive (i.e., parent selection), and how to update them (i.e., variation).

**Parent Selection.** Parent selection methods aim to select appropriate solutions to generate offspring solutions. Uniform random selection, i.e., selecting parent solutions from the archive uniformly at random, is one of the simplest selection methods and has been widely used in QD algorithms such as (Cully et al., 2015; Nilsson & Cully, 2021; Fontaine & Nikolaidis, 2021; Tjanaka et al., 2022). Novelty search with local competition (Lehman & Stanley, 2011) considers both the novelty score and local quality score (i.e., the number of neighbors that a solution outperforms), and selects parent solutions from the corresponding Pareto front. Curiosity (Cully & Demiris, 2018) calculates the curiosity score based on the history information (i.e., increases the score if the offspring solution improves the QD-Score; otherwise, decreases it), and selects the solutions with higher curiosity scores. EDO-CS (Wang et al., 2022) used a clustering-based selection method, which divides the archive into several clusters first and then selects good parent solutions from each cluster. NSS (Wang et al., 2023a) introduces a multi-objective optimization-based selection method based on surrounded dominance, and selects the solutions from the top-ranked fronts.

**Variation.** Vanilla ME uses the basic evolutionary operator, i.e., crossover and mutation, as the variation operator, which is sample-inefficient, especially in high-dimensional optimization problems. ME with Evolution Strategies (ME-ES) (Colas et al., 2020) and Covariance Matrix Adaptation-ME (CMA-ME) (Fontaine et al., 2020) use ES (Salimans et al., 2017) and CMA-ES (Hansen & Ostermeier, 2001), respectively, to make vanilla ME scalable. Policy Gradient Assisted-ME (PGA-ME) (Nilsson & Cully, 2021; Flageat et al., 2023a) employs policy gradient-assisted variation, which significantly improves sample efficiency compared to using only evolutionary operators. Objective and Measure Gradient ME via Gradient Arborescence (OMG-MEGA) (Fontaine & Nikolaidis, 2021) augments QD with explicit gradient information. QD-PG (Pierrot et al., 2022a) exploits information at the time-step level to promote QD search. Descriptor-Conditioned Gradients-ME (DCG-ME) (Faldor et al., 2023) improves policy gradient variation operator with a descriptor-conditioned critic that improves the archive across the entire descriptor space. Proximal Policy Gradient Arborescence

(PPGA) (Batra et al., 2024) adapts PPO to the DQD framework and enables efficient optimization and discovery of novel behaviors.

# B  DETAILS OF EXPERIMENTAL SETTINGS

## B.1  METHODS

We compare our implementation of CCQD with several baselines and state-of-the-art methods. For a fair comparison, we unify the common hyperparameters of these methods on all the eight environments. The other hyperparameters of each method are set as the corresponding original paper.

We represent a policy as a fully connected neural network with two 256-dimensional hidden layers, which is the same as (Chalumeau et al., 2023a) and can be represented as follows:

$$s \rightarrow \mathbf{MLP}(256) \rightarrow \mathbf{tanh} \rightarrow \mathbf{MLP}(256) \rightarrow \mathbf{tanh} \rightarrow \mathbf{MLP}(|\mathcal{A}|) \rightarrow \mathbf{tanh} \rightarrow a$$

where $s$ is the state, $a$ is the action, $|\mathcal{A}|$ is the dimension of the action space $\mathcal{A}$, and $\mathbf{MLP}(n)$ is a fully-connected layer with output size of $n$. All of the activation functions of the fully connected layers are $\tanh$.

Then, we introduce the two types of variation operators used in our experiments.

**IsoLineDD.**  IsoLineDD (Vassiliades & Mouret, 2018) is a popular evolutionary operator used in several QD algorithms (Nilsson & Cully, 2021; Grillotti et al., 2023; Chalumeau et al., 2023a; Lim et al., 2023a). Considering two parent solutions $\boldsymbol{x}_1$ and $\boldsymbol{x}_2$, the offspring solution $\boldsymbol{x}'$ generated by the IsoLineDD operator is sampled as follows:

$$\boldsymbol{x}' = \boldsymbol{x}_1 + \sigma_1 \mathcal{N}(\mathbf{0}, \mathbf{I}) + \sigma_2 (\boldsymbol{x}_2 - \boldsymbol{x}_1) \mathcal{N}(0, 1),$$

where $\sigma_1 = 0.005$ and $\sigma_2 = 0.05$ in this paper, $\mathbf{I}$ denotes the identity matrix, $\mathcal{N}(0, 1)$ and $\mathcal{N}(\mathbf{0}, \mathbf{I})$ denote a random number and a random vector sampled from the standard Gaussian distribution, respectively.

**Policy Gradient Operators.**  The policy gradient operator maintains a critic and a greedy actor in many QD algorithms (Nilsson & Cully, 2021; Pierrot et al., 2022a; Tjanaka et al., 2022; Chalumeau et al., 2023a; Lim et al., 2023a), and we adopt this setting as well. At the start of each generation, the critic is trained with TD loss, while the greedy actor is simultaneously trained with policy gradient. Subsequently, the parent solutions are updated with policy gradient, using the critic in the variation process. We employed the policy gradient method TD3, whose hyperparameters are presented in Table 2.

Table 2: The hyperparameters of TD3.

| Hyperparameter | Value |
|---|---|
| Critic hidden layer size | $[256, 256]$ |
| Policy learning rate | $1 \times 10^{-3}$ |
| Critic learning rate | $3 \times 10^{-4}$ |
| Replay buffer size | $1 \times 10^{6}$ |
| Training batch size | 256 |
| Policy training steps | 100 |
| Critic training steps | 300 |
| Reward scaling | 1.0 |
| Discount | 0.99 |
| Policy noise | 0.2 |
| Policy clip | 0.5 |

**Detailed Settings of Methods.**  The settings of the methods are summarized as follows.

- **ME** (Mouret & Clune, 2015) uses the IsoLineDD operator in the variation process, which is the same as (Chalumeau et al., 2023a).

- **QD-PG** (Pierrot et al., 2022a) uses the IsoLineDD operator, the policy gradient, and the diversity gradient in the variation process. The proportion of the offspring solutions generated by these three operators are all $1/3$.

- **PGA-ME** (Nilsson & Cully, 2021; Flageat et al., 2023a) uses the IsoLineDD operator, and the policy gradient in the variation process. The proportion of the offspring solutions generated by the two operators are both $0.5$, which is the same as (Nilsson & Cully, 2021; Lim et al., 2023a).

- **OMG-MEGA** (Fontaine & Nikolaidis, 2021; Tjanaka et al., 2022) uses the IsoLineDD and gradient ascent in the variation process. The gradient in OMG-MEGA is a random weighted sum of the normalized gradients of the fitness and the descriptor. In order to improve the sample efficiency of OMG-MEGA, we use TD3 to provide both of the gradients. The proportion of the offspring solutions generated by the two operators are both $0.5$, which is the same as PGA-ME.

- **PBT-ME** (Pierrot & Flajolet, 2023) evolves a population of agents instead of policies. In this paper, we use the SAC version of PBT-ME, which performs better than the TD3 version in the original paper.

- **CCQD** uses the variation operator as the same as PGA-ME. The representation population size $N_r$ is 20. We used (1+2) decomposition strategy in the experiments.

**Settings on Atari Pong.** For the *Atari Pong* (Bellemare et al., 2013) task with high-dimensional visual observation space and discrete action space, we change to use a DQN with three convolutional layers and two fully connected layers as our policy network, which is the same as (van Hasselt et al., 2016; Weng et al., 2022). All the other settings of CCQD and DQN-ME remain the same for a fair comparison, and the only one difference between CCQD and DQN-ME is the using of coevolution.

## B.2 DETAILS OF THE ENVIRONMENTS

The experiments of this paper are conducted on QDax[3] (Chalumeau et al., 2023b), a popular implementation of QD algorithms based on JAX[4], which contains several algorithms and environments (Lim et al., 2023a). The details of the environments are shown in Table 3, where $|\mathcal{S}|$ is the dimension of the state space $\mathcal{S}$, $|\mathcal{A}|$ is the dimension of the action space $\mathcal{A}$, $k$ is the dimension of the descriptor space, $|\mathcal{X}|$ is the number of parameters of the policy network, and #seeds is the number of random seeds used in each environment.

Table 3: The settings of the environments.

| Environments | $|\mathcal{S}|$ | $|\mathcal{A}|$ | $k$ | $|\mathcal{X}|$ | #seeds |
|---|---|---|---|---|---|
| *Hopper Uni* | 11 | 3 | 1 | $69,635$ | 10 |
| *Walker2D Uni* | 17 | 6 | 2 | $71,942$ | 5 |
| *HalfCheetah Uni* | 18 | 6 | 2 | $72,198$ | 5 |
| *Ant Uni* | 87 | 8 | 4 | $90,376$ | 5 |
| *Humanoid Uni* | 227 | 17 | 2 | $128,529$ | 5 |
| *Humanoid Omni* | 227 | 17 | 2 | $128,529$ | 10 |
| *Point Maze* | 2 | 2 | 2 | $67,074$ | 5 |
| *Ant Maze* | 101 | 8 | 2 | $93,960$ | 10 |

## B.3 COMPUTATIONAL RESOURCES

The experiments are conducted on an NVIDIA RTX 3090 GPU (24 GB) with an AMD Ryzen 9 3950X CPU (16 Cores), except for PBT-ME, which is conducted on an NVIDIA RTX A6000 GPU (48 GB) with an AMD EPYC 7763 CPU (64 Cores).

---

[3]https://github.com/adaptive-intelligent-robotics/QDax
[4]https://github.com/google/jax

## C ADDITIONAL RESULTS

### C.1 QD-SCORE TABLE

QD-Score focuses more on the overall quality of the final obtained archive, rather than on the efficiency of an algorithm as in the case of QD-Score AUC. As shown in Table 4, CCQD achieves the best QD-Scores across all environments and has the highest rank. Besides, CCQD is significantly better than all the other algorithms in seven experimental environments, while its performance on *Ant Maze* is almost equivalent to that of QD-PG and PGA-ME. Compared to QD-Score AUC (i.e., Table 1), the overall rankings are consistent, except for an exchange in the ranking order of QD-PG and OMG-MEGA.

Table 4: QD-Score of different methods on eight environments with episode length 250 and total timesteps $5e7$. The symbols '+', '−', and '≈' indicate that the result is significantly superior to, inferior to, and almost equivalent to CCQD, respectively, according to the Wilcoxon rank-sum test with significance level 0.05. **Bold** and underline texts respectively denote the best and runner-up algorithms.

| Environment | ME | QD-PG | PGA-ME | OMG-MEGA | PBT-ME | CCQD |
|---|---|---|---|---|---|---|
| *Hopper Uni* | 559415.3 − | 507988.9 − | 616474.4 − | 607255.4 − | 521933.1 − | **646739.6** |
| *Walker2D Uni* | 685905.1 − | 689118.9 − | 741324.0 − | 740915.6 − | 518859.5 − | **783199.7** |
| *HalfCheetah Uni* | 2274903.9 − | 2188888.8 − | 2580297.8 − | 2625036.0 − | 2734491.6 − | **2865478.6** |
| *Ant Uni* | 804234.5 − | 862769.3 − | 868586.4 − | 894949.9 − | 671408.9 − | **931191.1** |
| *Humanoid Uni* | 791793.6 − | 825267.3 − | 751922.0 − | 746152.3 − | 606250.8 − | **877268.5** |
| *Humanoid Omni* | 5886.8 − | 8618.0 − | 7813.4 − | 7022.7 − | 7031.5 − | **15063.2** |
| *Point Maze* | 263770.7 − | 278524.3 − | 233201.7 − | 232718.3 − | 176060.9 − | **377520.2** |
| *Ant Maze* | 695519.4 − | 1023962.7 ≈ | 972198.9 ≈ | 972772.0 − | 788371.8 − | **1038160.7** |
| $+/-/\approx$ | 0/8/0 | 0/7/1 | 0/7/1 | 0/8/0 | 0/8/0 | / |
| Average Rank | 4.62 | 3.50 | 3.25 | 3.62 | 5.00 | **1.00** |

### C.2 ARCHIVE PROFILE

**Archive Profile** (Flageat et al., 2022) denotes the number of solutions in the archive whose quality is better than a threshold. It can measure the quality of the final obtained archive. We additionally plot the archive profile (Flageat et al., 2022) to evaluate the quality of the final archives of different methods, as shown in Figure 5. The archive profile shows the number of solutions in the archive whose quality is better than a threshold. As the threshold increases, CCQD exhibits the slowest decay in the number of solutions across multiple environments. This highlights the superior quality of CCQD's final archive.

### C.3 CORRECTED METRICS

The metrics in the noisy (or uncertain) setting are important for downstream tasks. Here, we considered the noisy setting, by reevaluating the solution of each cell in the archive for 50 times and filling a new archive called "Corrected Archive" (Grillotti et al., 2023; Flageat et al., 2023b; Flageat & Cully, 2023) according to the average fitness and behavior descriptors. We use the corresponding QD metrics of the Corrected Archive, i.e., Corrected QD-Score, Corrected Coverage, and Corrected Max Fitness, which are popular for measuring the performance of QD algorithms in the noisy environments. As shown in Table 5, CCQD still outperforms PGA-ME in all environments.

### C.4 NUMBER OF REPRESENTATION POPULATION SIZE.

Table 6 gives the QD-Score obtained by PGA-ME as well as CCQD with different representation population size $N_r$. We test $N_r = 1, 10, 20, 50$. The last row also shows the average sample efficiency improvement of CCQD over PGA-ME, when reaching the QD-Score of PGA-ME, on the eight environments. We can observe that equipped with different values of $N_r$, CCQD is always better than PGA-ME and can improve the sample efficiency significantly. However, choosing an inappropriate value for $N_r$ will also decrease the improvement. If $N_r$ is too small, it will oversimplify

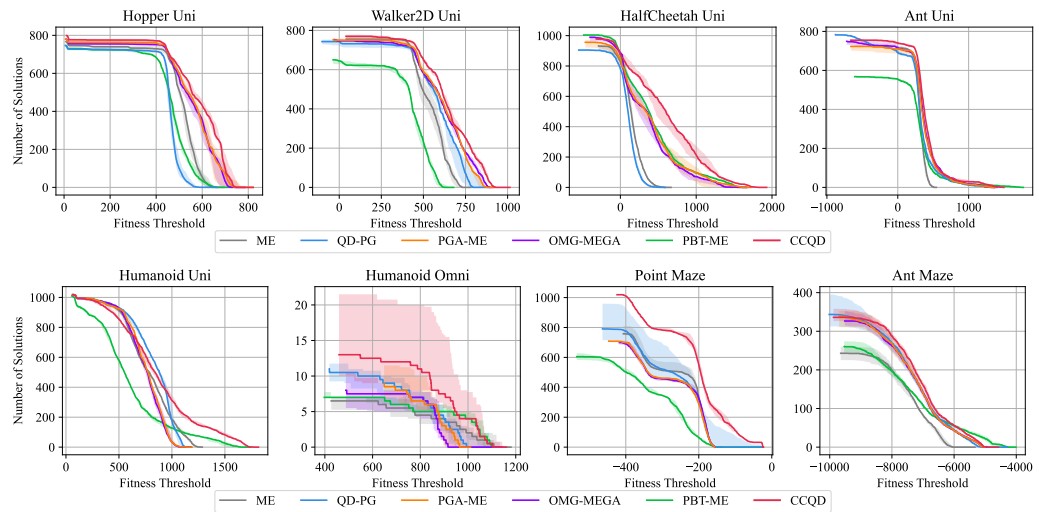

Figure 5: The archive profile of different methods on eight environments with episode length 250 and total timesteps $5e7$. The medians and the first and third quartile intervals are depicted with lines and shaded areas, respectively.

Table 5: Corrected QD-Score, Corrected Coverage, and Corrected Max Fitness of PGA-ME and CCQD on different environments with episode length 250 and total timesteps $5e7$. **Bold** text respectively denote the better algorithm.

|  | Corrected QD-Score | | Corrected Coverage | | Corrected Max Fitness | |
|---|---|---|---|---|---|---|
|  | PGA-ME | CCQD | PGA-ME | CCQD | PGA-ME | CCQD |
| *HalfCheetah Uni* | 876170.00 | **1233301.75** | 34.38 | **43.85** | 1162.18 | **1361.92** |
| *Ant Uni* | 330165.94 | **403013.94** | 28.12 | **32.23** | 985.80 | **1151.37** |
| *Humanoid Uni* | 245687.56 | **337272.06** | 51.37 | **52.83** | 807.86 | **1625.46** |
| *Point Maze* | 140449.19 | **221739.34** | 40.92 | **58.50** | -156.55 | **-24.52** |

Table 6: QD-Score of PGA-ME and CCQD with different representation population sizes on eight environments with episode length 250 and total timesteps $5e7$. The last row indicates the average percentage of sample efficiency improvement achieved by various variants of CCQD, when reaching the QD-Score of PGA-ME.

| Environment | PGA-ME | CCQD-1 | CCQD-10 | CCQD-20 | CCQD-50 |
|---|---|---|---|---|---|
| *Hopper Uni* | 616474.4 | 658986.5 | 656077.2 | 646739.6 | 631475.0 |
| *Walker2D Uni* | 741324.0 | 726144.0 | 775977.0 | 783199.7 | 783983.6 |
| *HalfCheetah Uni* | 2580297.8 | 2619269.3 | 2764638.6 | 2865478.6 | 2711417.7 |
| *Ant Uni* | 868586.4 | 869856.9 | 913514.0 | 931191.1 | 892193.8 |
| *Humanoid Uni* | 751922.0 | 899925.4 | 895138.7 | 877268.5 | 941327.0 |
| *Humanoid Omni* | 7813.4 | 14450.3 | 15678.7 | 15063.2 | 19328.3 |
| *Point Maze* | 233201.7 | 253359.4 | 318775.7 | 377520.2 | 274164.5 |
| *Ant Maze* | 972198.9 | 828489.4 | 980763.7 | 1038160.7 | 979034.6 |
| Average Rank | 4.75 | 3.62 | 2.25 | 2.00 | 2.38 |
| Average % of Improvements | / | 165.3 | 230.0 | 246.6 | 217.0 |

the problem, while a too large value will fail to fully demonstrate the advantage of CCQD. Therefore, we recommend using a moderate value for $N_r$, such as 10 or 20. We used 20 in our experiments.

## C.5 NUMBER OF CRITICS

We conduct an ablation study on the number of critics, including CCQD (single TD3 critic), and CCQD (single PeVFA critic). As shown in Figure 6, CCQD we used is the best, validating the effectiveness of maintaining multiple PeVFAs. Besides, CCQD (single TD3 critic) and CCQD (single PeVFA) both outperform PGA-ME, indicating that they are still sample-efficient and can be used in some scenarios that have limited memory resources.

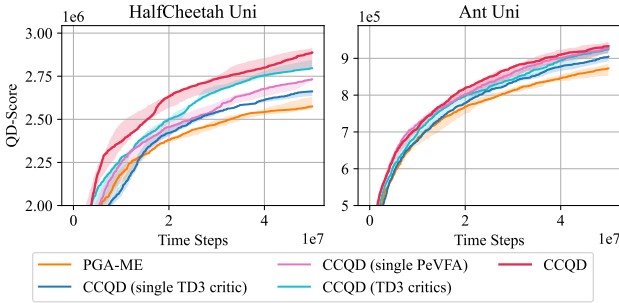

Figure 6: The ablation study on the number of critics.

## C.6 ABLATION STUDY ON PEVFA

In our implementation of CCQD, we utilized PeVFA as our critic. Here, we conduct an ablation study to compare its performance with the implementation without PeVFA, i.e., using the critic of TD3. As shown in Figure 7, both CCQD and CCQD (without PeVFA) exhibit significantly better performance than PGA-ME. Moreover, using PeVFA as the critic may result in relatively higher performance.

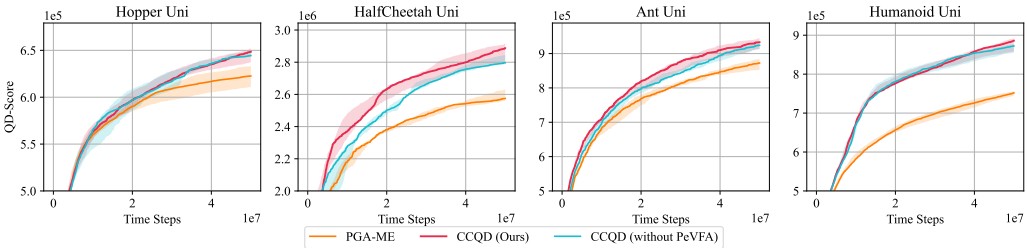

Figure 7: The ablation study on PeVFA.

## C.7 VISUALIZATION

**Final archive.** To clearly show the (decision) archive quality, we visualize the archives obtained by ME, PGA-ME, and CCQD. As shown in Figure 8(a)–(c), the qualities of the solutions in the archive obtained by CCQD are much higher than ME and PGA-ME, and CCQD also covers more regions of the archive. In addition, we want to know if the representation archive is really diverse. For each decision part in the final decision archive, we combine it with all the representation parts in the representation archive, evaluate them, and calculate the standard derivation of the fitness, as shown in Figure 8(d). We can observe that the standard deviations in many regions are large, reflecting the good abilities of the representation parts in recognizing diverse state representations.

**Running process of *Point Maze*.** *Point Maze* is a maze environment that involves walking from the green start point in the lower right corner to the red end point in the upper left corner, as shown in Figure 9(a). The reward is related to the distance from the end point, and the closer to the end point the higher the reward. We can observe from Figure 9(b)-(e) that the algorithm bypasses the wall in the middle of the training process; thus, it comes closer to the end point and quickly explores the neighborhood, making all the metrics jump significantly.

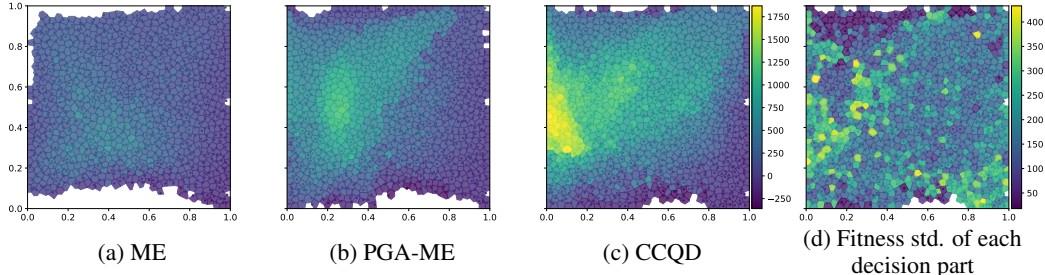

| (a) ME | (b) PGA-ME | (c) CCQD | (d) Fitness std. of each decision part |

Figure 8: (a)-(c): Visualization of the archives obtained by ME, PGA-ME, and CCQD on *HalfCheetah Uni*, where the 2-dimensional behavior space is discretized into cells. (d): Standard derivation of fitness of each decision part, when combined with different representation parts, of CCQD on *HalfCheetah Uni*. Note that colors represent the quality in (a)-(c), and standard deviation in (d).

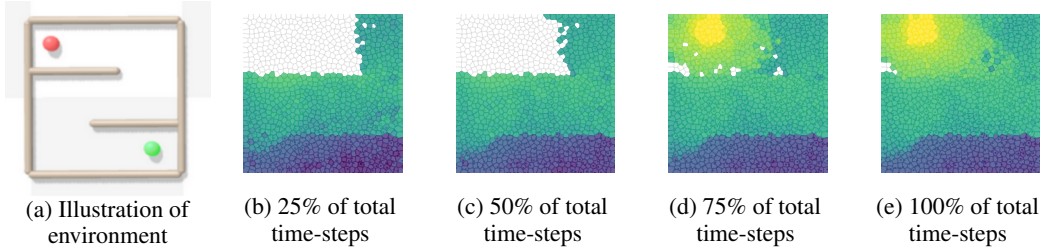

| (a) Illustration of environment | (b) 25% of total time-steps | (c) 50% of total time-steps | (d) 75% of total time-steps | (e) 100% of total time-steps |

Figure 9: (a): Illustration of *Point Maze* environment (Chalumeau et al., 2023a). The green dot represents the starting position, and the red dot represents the target position. (b)–(e): Visualization of the archives on *Point Maze* in the different stages of the training process, where the 2-dimensional behavior space is discretized into cells, and the colors represent the fitness.

**Internal dynamics of decision population.** We analyze the proportion of surviving IsoLineDD solutions and surviving RL solutions to all surviving solutions in each generation. We plot the curves of the proportion of RL solutions on Hopper Uni and Walker2D Uni in Figure 10, and smooth them for better presentation. Note that the proportion of IsoLineDD solutions is (1 - proportion of RL solutions). We can observe that as the optimization proceeds, the proportion of RL solutions tends to decrease, which is consistent with the results of PGA-ME (see Figure 10 in (Flageat et al., 2023a)). This may be because the IsoLineDD operator can find diverse solutions throughout the whole process, due to its strong capability of global exploration.

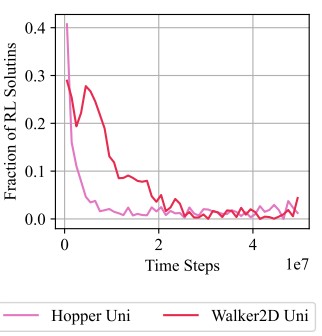

Figure 10: The fraction of the solutions reproduced by the RL operator among the surviving offspring.

### C.8 OTHER SETTINGS

**AURORA** We also compare CCQD and PGA by adopting AURORA (Grillotti & Cully, 2022), denoted as CCQD-AURORA and PGA-AURORA. As shown in Table 7, CCQD-AURORA archives better QD-Scores across all environments, demonstrating the generalization of CCQD.

**Longer Episode length** We also compare all the methods in the environments with episode length 1000, which is consistent with the setting in Pierrot & Flajolet (2023). As shown in Table 8 and Figure 11, the Max Fitness of the methods in these environments now is about 4 times as high as the environments with episode length 250, and is consistent with the results in the previous work that use 1000 as the episode length, such as Pierrot & Flajolet (2023). In this setting, CCQD still has the best average rank on QD-Score AUC.

Table 7: QD-Score, Coverage, and Max Fitness of different methods on two environments with episode length 250 and total timesteps $2e7$. **Bold** texts respectively denote the best algorithms.

| | QD-Score | | Coverage | | Max Fitness | |
|---|---|---|---|---|---|---|
| | PGA-AUR. | CCQD-AUR. | PGA-AUR. | CCQD-AUR. | PGA-AUR. | CCQD-AUR. |
| *Hopper Uni* | 122789.1 | **135967.8** | 37.9 | **48.5** | **642.9** | 529.0 |
| *Walker2D Uni* | 154533.4 | **181841.0** | 38.7 | **45.0** | 655.6 | **775.2** |

Table 8: QD-Score AUC ($\times 10^{12}$) of different methods on eight environments with episode length 1000 and total timesteps $1.5e8$. The symbols '+', '−', and '≈' indicate that the result is significantly superior to, inferior to, and almost equivalent to CCQD, respectively, according to the Wilcoxon rank-sum test with significance level 0.05. **Bold** and underline texts respectively denote the best and runner-up algorithms.

| Environment | ME | QD-PG | PGA-ME | OMG-MEGA | PBT-ME | CCQD |
|---|---|---|---|---|---|---|
| *Hopper Uni* | 224.65 − | 178.45 − | 253.63 − | 244.79 − | 226.14 − | **264.03** |
| *Walker2D Uni* | 248.72 − | 289.76 − | 313.38 − | 314.58 − | 234.59 − | **387.04** |
| *HalfCheetah Uni* | 1252.82 − | 1104.65 − | 1304.95 − | 1329.01 − | **1624.75** + | 1484.76 |
| *Ant Uni* | 374.50 − | 423.53 − | 426.05 − | 439.96 − | **477.42** ≈ | 468.48 |
| *Humanoid Uni* | 117.43 − | 156.25 ≈ | 127.73 − | 131.78 − | 108.48 − | **170.81** |
| *Humanoid Omni* | 0.90 − | 4.68 ≈ | 11.05 ≈ | 12.59 ≈ | 1.42 − | **20.32** |
| *Point Maze* | 127.83 ≈ | 125.45 ≈ | 121.72 ≈ | 117.95 − | **131.42** ≈ | 130.88 |
| *Ant Maze* | **1038.69** ≈ | 1015.63 ≈ | 1018.56 ≈ | 961.73 ≈ | 940.29 ≈ | 988.62 |
| +/ − / ≈ | 0/6/2 | 0/4/4 | 0/5/3 | 0/6/2 | 1/4/3 | / |
| Average Rank | 4.50 | 4.25 | 3.38 | 3.38 | 3.75 | **1.75** |

Table 9: Max Fitness of different methods on eight environments with episode length 250 and total timesteps $1.5e8$. The symbols '+', '−', and '≈' indicate that the result is significantly superior to, inferior to, and almost equivalent to CCQD, respectively, according to the Wilcoxon rank-sum test with significance level 0.05. **Bold** and underline texts respectively denote the best and runner-up algorithms.

| Environment | ME | QD-PG | PGA-ME | OMG-MEGA | PBT-ME | CCQD |
|---|---|---|---|---|---|---|
| *Hopper Uni* | 697.9 − | 634.8 − | **820.5** ≈ | 806.8 ≈ | 702.1 − | 788.1 |
| *Walker2D Uni* | 788.5 − | 842.3 − | 902.9 − | 920.9 − | 692.3 − | **1003.8** |
| *HalfCheetah Uni* | 723.2 − | 584.9 − | 1830.7 ≈ | 1801.4 ≈ | 1767.9 − | **1920.4** |
| *Ant Uni* | 618.6 − | 1427.2 − | 1458.3 − | 1492.5 − | **1752.0** ≈ | 1688.6 |
| *Humanoid Uni* | 1284.2 − | 1293.0 − | 1397.7 − | 1548.2 − | 1563.6 ≈ | **1796.3** |
| *Humanoid Omni* | 1096.0 + | 1006.8 ≈ | 998.4 ≈ | 955.9 ≈ | **1133.4** + | 999.9 |
| *Point Maze* | −55.8 − | −76.7 ≈ | −153.2 − | −156.6 − | −105.0 − | **-23.6** |
| *Ant Maze* | −5643.3 − | −4835.2 − | −4822.9 − | −4768.0 − | **-3932.8** + | −4572.0 |
| +/ − / ≈ | 1/7/0 | 0/6/2 | 0/5/3 | 0/5/3 | 2/4/2 | / |
| Average Rank | 4.62 | 4.62 | 3.50 | 3.50 | 2.88 | **1.88** |

**Max Fitness Tables** Max Fitness can measure the exploitation ability of a QD algorithm. As shown in Table 9-10, CCQD has the best average rank in both environment settings, and PBT-ME is the runner-up.

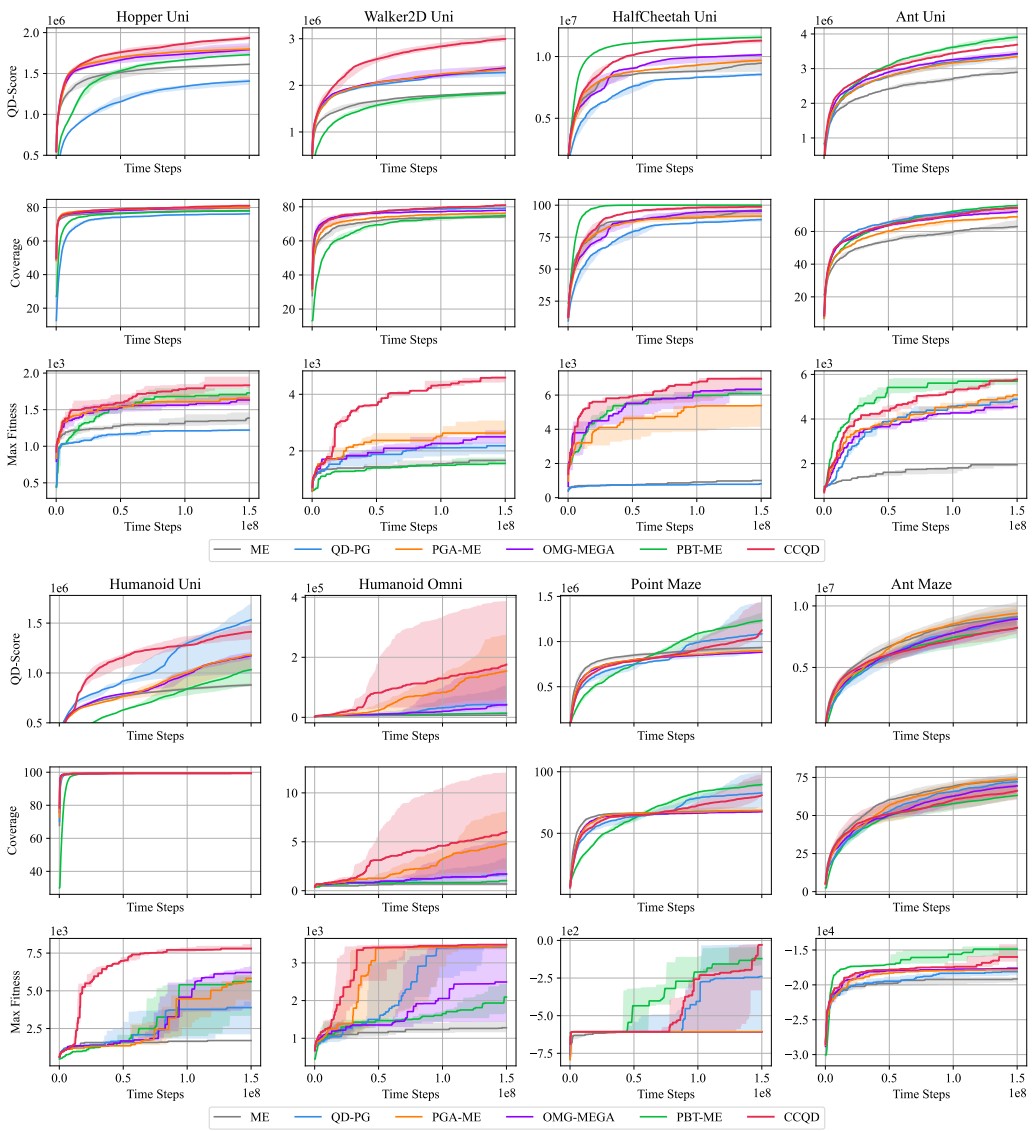

Figure 11: Performance comparison of CCQD with other methods in terms of QD-Score, Coverage, and Max Fitness on eight environments with episode length $1000$ and total timesteps $1.5e8$. The medians and the first and third quartile intervals are depicted with curves and shaded areas, respectively.

Table 10: Max Fitness of different methods on eight environments with episode length $1000$ and total timesteps $1.5e8$. The symbols '$+$', '$-$', and '$\approx$' indicate that the result is significantly superior to, inferior to, and almost equivalent to CCQD, respectively, according to the Wilcoxon rank-sum test with significance level 0.05. **Bold** and underline texts respectively denote the best and runner-up algorithms.

| Environment | ME | QD-PG | PGA-ME | OMG-MEGA | PBT-ME | CCQD |
|---|---|---|---|---|---|---|
| *Hopper Uni* | $1376.4\,-$ | $1222.9\,-$ | $1673.9\,-$ | $1644.8\,-$ | $\underline{1752.8}\approx$ | **1878.2** |
| *Walker2D Uni* | $1657.6\,-$ | $2132.0\,-$ | $\underline{2763.2}\,-$ | $2509.0\,-$ | $1639.8\,-$ | **4534.7** |
| *HalfCheetah Uni* | $1008.3\,-$ | $828.8\,-$ | $5033.9\,-$ | $5933.9\approx$ | $\underline{6225.1}\approx$ | **6643.0** |
| *Ant Uni* | $1888.2\,-$ | $4880.4\,-$ | $4935.4\,-$ | $4611.3\,-$ | **5795.6**$\approx$ | $\underline{5741.0}$ |
| *Humanoid Uni* | $1724.2\,-$ | $3689.9\,-$ | $5268.3\,-$ | $\underline{5355.2}\,-$ | $4553.4\,-$ | **7858.2** |
| *Humanoid Omni* | $1310.4\,-$ | $3146.0\approx$ | $\underline{3241.1}\approx$ | $2441.7\approx$ | $2054.4\,-$ | **3278.2** |
| *Point Maze* | $-582.7\,-$ | $-301.4\approx$ | $-585.2\,-$ | $-606.7\,-$ | **-130.1**$\approx$ | $\underline{-168.4}$ |
| *Ant Maze* | $-19220.2\,-$ | $-18119.1\,-$ | $-17769.6\,-$ | $-17620.9\,-$ | **-15260.7**$\approx$ | $\underline{-15538.1}$ |
| $+/-/\approx$ | 0/8/0 | 0/6/2 | 0/7/1 | 0/6/2 | 0/3/5 | / |
| Average Rank | 5.38 | 4.50 | 3.25 | 3.75 | 2.75 | **1.38** |

