# OpenReview forum: "Sample-Efficient Quality-Diversity by Cooperative Coevolution"
_ICLR.cc/2024/Conference — ICLR 2024 spotlight_

### Official Review · Reviewer_VT9L · 2023-10-30

**Soundness:** 2 fair
**Presentation:** 2 fair
**Contribution:** 3 good
**Rating:** 8
**Confidence:** 5

**Summary:**

This paper proposes the cooperative coevolution for QD methods on reinforcement learning problems. The major contribution is claimed to be the first time use of cooperative coevolution, and this can help improve the sample efficiency.

**Strengths:**

The experiment setup is considerate, and the results are promising. The main idea of using cooperative coevolution into reinforcement learning is easy to understand for addressing large-scale search space.

**Weaknesses:**

This work is not the first time of applying cooperative coevolution to reinforcement learning, even restricted in the quality-diversity like evolutionary algorithms, see the following reference:
[1] Evolutionary reinforcement learning via cooperative coevolutionary negatively correlated search.
So the related claims should be revised.
Given the above work, the contribution of this work may be limited.

**Questions:**

What is the relationship between large-scale search space and the sample efficiency? Why address the large-scale space with cooperative coevolution can help improve the sample efficiency of QD?

---

> ### Author Response · Authors · 2023-11-20
> **Response by the authors**
>
> Thanks for your comments. Below please find our responses.
>
> ## Q1 This work is not the first time of applying cooperative coevolution to reinforcement learning. Contribution is limited.
>
> Thank you for bringing up the paper [1], which is an interesting work proposing an efficient method called Cooperative Coevolutionary Negatively Correlated Search (CCNCS) that utilizes CC to solve large-scale optimization problems in RL.
>
> We would like to clarify that our main contribution is not the initial application of CC to RL, but rather the incorporation of CC principles into QD, along with the introduction of a sample-efficient CCQD framework.
>
> Apart from the motivation and application tasks, the differences in the method between our work and [1] are significant, which are explained as follows:
>
> 1. Decomposition strategy. Inspired by recent advancements in representation learning, CCQD employs a layer-based decomposition strategy and divides the policy network into two parts with different functions. CCNCS utilizes the random grouping strategy, which is commonly used in CCEA.
>
> 2. Population size and shared components. CCQD not only employs a layer-based decomposition strategy but also maintains fewer representation parts. This means that the population sizes for the two sub-problems in CCQD are different, which distinguishes it from many traditional CC approaches, including [1]. This unique characteristic of CCQD is derived from a specific observation of the QD task: maintaining smaller numbers of representation parts and sharing them further serves our goal of simplifying the optimization problem. CCNCS does not share the same part due to its parallel exploration behaviors.
>
> We have included a discussion about this work in our main paper. Thank you.
>
> ## Q2 Relationship between large-scale search space and the sample efficiency? Why address the large-scale space with cooperative coevolution can help improve the sample efficiency of QD?
>
> We consider the problem of QDRL and optimize the parameters of the policy network by sampling from the environment. Therefore, the number of samples used can be viewed as the number of evaluations in a typical optimization problem. In the case of a large-scale optimization problem, if the search space can be appropriately reduced, a smaller number of evaluations can be used to find satisfactory solutions. CCQD achieves this by decomposing the problem and using shared representation, effectively reducing the optimization space. Additionally, through cooperative coevolution, the collaboration between the representation part and the decision part promotes exploration and enhances the ability to find diverse policies. As a result, CCQD improves the optimization efficiency, i.e., the sample efficiency of QD.
>
> [1] Evolutionary reinforcement learning via cooperative coevolutionary negatively correlated search. Swarm and Evolutionary Computation, 2022.
>
> **We hope that our response has addressed your concerns, but if we missed anything please let us know.**

---

> > ### Comment · Reviewer_VT9L · 2023-11-21
> > **Good clarification**
> >
> > Thank the authors for detailed clarification, which makes the contribution clear to me. I think this work makes a good contribution, which utilizes the idea of cooperative coevolution to improve the sample efficiency of QD and may encourage more works on the combination of these two topics. I have changed my recommendation to “accept”.

---

### Official Review · Reviewer_CbDL · 2023-10-31

**Soundness:** 2 fair
**Presentation:** 3 good
**Contribution:** 3 good
**Rating:** 8
**Confidence:** 5

**Summary:**

In that paper, the authors introduce Cooperative Coevolution QD (CCQD), a novel method to evolve populations of diverse and high-performing RL agents, the so-called QDRL problem. The paper aims to improve the sample efficiency of current methods by disentangling the feature extraction learning, also called representation learning part and the decision learning part of the policies during training. This done by introducing two separate populations that are co-evolved with the representation population being much smaller than the decision one. This is motivated by the intuition that most of the representation knowledge can be re-used by all the agents while the diversity and performance is more likely to emerge from the decision parts. The authors propose to split the policy networks layers to define these populations where the first group of layer and the last group of layers would respectively define the representation and decision populations.

The framework is general and most of the design choices used by the authors are common in that literature including the type of archives and variation operators. The authors introduced two changes compared to others such as PGA-MAP-Elites in the way the critics are handled.

Their method is then benchmarked on the QDax suite against several state-of-the-art methods (PGA-ME, QDPG, ME and PBT-ME (SAC)).

**Strengths:**

Overall I find that this paper addresses an interesting topic in the QDRL literature which is well motivated. The approach is sound and I share the same intuition that these methods would strongly benefit from different treatments of the representation learning and decision learning parts. Using co-evolution to do so is novel and I find it also elegant.

The paper is overall well written and does a good job at setting up the context, the problem and presents a good overview of the literature on this topic. The technical aspects are also well explained and I found it easy to understand how the algorithm works and is implemented.

Regarding the experimental section, I find the ablation study informative and I like the introduction of the Atari Pong environment which is not very common in that field but I think very useful to show the versatility of the approaches.

**Weaknesses:**

However, I have strong doubts about the larger benchmark where CCQD is compared on the QDax suite to QDRL competitors. While I appreciate that the authors considered a large number of environments and repeated each experiment five times, I have serious doubts about the showed result as they don't match the ones reported in recent works such as PBT-ME (ICLR last year) produced on the exact same benchmark and for which the code is available.

- My main concern is the number of time steps considered and the maximum fitness obtained in several environments.  For instance in Ant-Uni, the authors show training evolution over 5e7 time steps for a maximum fitness reached of 1500. This fitness corresponds to the robot barely moving at low speed. In that environment, asymptotic fitnesses are more in the scale of 6000-7000, see for instance PBT-ME, which corresponds to the ant robot actually running. Same observation in HalfCheetah where the authors report asymptotic performance of 1500 while usual values are around 6000-7000 too. It is even worse in Walker2d-Uni where reported performance is 800 compared to 4000 in state-of-the-art works. This indicates that the shown results correspond to very early stages of training, before actual convergence, and thus cannot allow to draw any meaningful conclusion.

- The authors claim "PBT-ME is capable of adjusting the algorithm’s hyperparameters while optimizing, resulting in the need of a significant amount of computational resources" while of the main point and contribution of the paper is to show that tuning the hyperparameters can be done with the same budget, and even in some cases improve the efficiency. The reported conclusions and performance ranking to other methods are not in the line with the one published while both the code and benchmark are open. I think this might come from the fact that all the runs were stopped too early and I would expect the authors to find back the original paper claim if they wait long enough to see the beginning of training convergence.

- Moreover, despite these limitations, in most cases the sample efficiency gain that is shown remains limited and I would expect more from an efficient representation/decision decoupling. In that context it is also hard to say if the small obtained gains compared to PGA-ME for instance are due to the co-evolution strategy being proposed or to some other minor changes like the PeVFA update of the critic or the fact that the population critic has been replaced by one critic per agent.

All in all, I am happy to see such works submitted at ICLR and hope there will be more in the future as I believe QDRL is a very exciting field and might play an important role in the future of AI methods. I also like the motivation and find the method original. The paper is also pleasant to read. However, as per my points above, I think that the main experiments of the paper show strong weaknesses which makes it unconvincing. I would be happy if the authors could mitigate these points by notably running all methods longer, until observing the beginning of asymptotic convergence, and also by confirming findings of previous works. In case the authors find different results of previous works, I would expect a strong discussion about what in their opinion could trigger these changes. If the authors were able to strengthen their experimental setup I would be happy to see this work published at ICLR but in the current state I think the work is not ready.

**Questions:**

**Minor points**:

- (manon et al.) ---> (flageat et al.) page 7

- It might be interesting to cite "The Quality-Diversity Transformer: Generating Behavior-Conditioned Trajectories with Decision Transformers" as the work is recent and was published at GECCO and here is also this idea of compressing the feature extraction within a single model, even though both works share different final motivations.

- I would have liked to see more discussion about the co-evolution aspect of the method, the different existing strategies and the motivation behind the design choices (e.g. the split of the policy layers). Maybe it would be worth it to move part of the other technical details that are common to most QDRL works to the supplementary to focus more in the main core on the aspects that make this work unique.

---

> ### Author Response · Authors · 2023-11-20
> **Response by the authors (1/2)**
>
> Thank you for your constructive and valuable comments. Below please find our response.
>
>
> ## Q1 Longer length and inconsistent with previous works
>
> Thanks for your suggestion. As this work is more concerned with the sample efficiency, we originally trained over 5e7 timesteps. According to your suggestion, we have revised to run all the methods longer to show their asymptotic performance. As shown in Table 1,  Figure 2, and Table 8, our main conclusion still holds. The average ranks of the methods do not change a lot, where the ranks of QD-PG and PBT-ME get better slightly, and those of others get worse slightly. CCQD still has the best average rank (i.e., 1.12 and 1.88, respectively) on QD-Score AUC and Max Fitness.
>
> Thanks to your suggestion, we have checked the reason for the roughly 4-times difference (e.g., 1500 vs. 6000) in Max Fitness that you mentioned. We find that the difference is not due to the lack of convergence, but to the different settings of the environments. We used episode length 250 in our initial experiments, which is consistent with previous works [1-2] and the default setting of QDax [3], while the maximum episode length is 1000 in [4]. The fitness of the environments is mainly determined by the total distance the robot moves, and a longer episode length allows the robot to move farther. Thus, the 4-times maximum episode length of the environments results in approximately 4-times Max Fitness. The similar results with our paper in these environments can be found in previous works [1-2].
>
> To further illustrate this, we additionally re-run all the methods in the environments with episode length 1000, which is consistent with the setting in [4]. As shown in Table 7, Table 9 and Figure 11, the Max Fitness of the methods in these environments now is about 4 times as high as our original environments, and is consistent with the results in the previous work that use 1000 as the episode length, such as [4]. In this setting, CCQD still has the best average rank (i.e., 1.75 and 1.38, respectively) on QD-Score AUC and Max Fitness.
>
> From these updated and newly added empirical results, we can also find that the performance rank of PBT-ME with previous methods is mostly consistent with that reported in the original paper of PBT-ME [4]. For example, in Figure 11, the QD-Scores of PBT-ME are better than those of PGA-ME and QD-PG in Ant Uni and HalfCheetah Uni, and worse than those in Walker2D Uni; in Tables 8-9 (which give the comparison of the Max Fitness of different methods and are added to the revised paper due to your suggestion) in Appendix C, PBT-ME achieves the highest average rank of Max Fitness except CCQD in both environmental settings.
>
> We put the added results of using the episode length 1000 in the appendix. But if you think that it would be beneficial to present the results of using the episode length 1000 rather than 250 in the main paper, please feel free to let us know.
>
>
> ## Q2 Different performance of PBT-ME
>
> The implementation of PBT-ME is based on its official implementation in QDax [5]. In our initial experiments, we kept the common hyperparameters consistent (e.g., population size and variation operators) to ensure a fair comparison. Thanks to your comment, we have revised to utilize the setting from the original paper to re-run PBT-ME. Here is the PBT-ME's performance comparison between our setting and the original setting:
>
> |                 |    QD-Score |  QD-Score |    Coverage | Coverage | Max Fitness | Max Fitness |
> |-----------------|------------:|----------:|------------:|---------:|------------:|------------:|
> |                 | Our Setting |  Original | Our Setting | Original | Our Setting |    Original |
> | Hopper Uni      |    609244.1 |  609217.0 |        75.8 |     77.3 |       770.6 |       702.1 |
> | Walker2D Uni    |    633808.3 |  654797.5 |        69.6 |     72.7 |       725.1 |       692.3 |
> | HalfCheetah Uni |   2974148.7 | 3020206.8 |       100.0 |    100.0 |      1928.4 |      1767.9 |
> | Ant Uni         |    854323.4 |  952361.1 |        67.4 |     73.3 |      1808.1 |      1752.0 |
> | Point Maze      |    234876.8 |  283454.1 |        73.3 |     81.5 |      -131.0 |      -105.0 |
> | Ant Maze        |   1090822.0 | 1111988.9 |        32.4 |     34.5 |     -3498.6 |     -3932.8 |
>
> We find that the results of PBT-ME under the two settings are quite similar. Under our setting, the Max Fitness is slightly higher, while under the original setting, the QD-Score is slightly higher. Note that we have not re-run PBT-ME in the two humanoid-based environments due to the limitation of device; we will do it and update the results in the final version.

---

> ### Author Response · Authors · 2023-11-20
> **Response by the authors (2/2)**
>
> ## Q3 Performance Gain by coevolution
>
> Thanks for your comments. We believe the performance gain by coevolution is significant. The reasons are as follows.
>
> 1. The main difference between CCQD and DQN-ME in the context of Atari-Pong is the utilization of coevolution, where CCQD does not use other techniques such as PeVFA. We can find from Figure 4(c) that the performance of CCQD is significantly better than DQN-ME in Atari-Pong, which thus validates the effectiveness of the coevolution strategy. We have made revisions to the paper to enhance the clarity of this point. Thank you for your feedback.
>
> 2. We conducted ablation studies on the number of critics and using of PeVFA, as shown in Appendix C.6 and C.7, respectively. The results show that using only one critic or not employing PeVFA does not decrease the performance significantly. Moreover, the performance remains significantly superior to the PGA-ME baseline, highlighting the effectiveness of coevolution.
>
>
> ## Q4 Missing one related work
>
> Thanks for pointing this out. We have revised to add some discussion in the new version.
>
> In the Related Work part:
> "The Quality-Diversity Transformer (QDT) [6] compresses an entire archive into a single behavior-conditioning policy, which can be used for many downstream applications."
>
> In the Conclusion part:
> "Additionally, the shared representation components in CCQD offer the potential for reducing the storage cost of QD algorithms. In some computationally-constrained application tasks, combining the CCQD framework with archive distillation techniques [6] can further reduce the storage cost and achieve a sample and storage-efficient QD algorithm, which is an aspect we will investigate in future research."
>
>
> ## Q5 More discussions about the co-evolution
>
> According to your suggestion, we have revised to move some technical details that are common to QDRL in Section 2.3 to the appendix and add more discussions about the co-evolution. We hope it is clearer now. Thank you.
>
> [1] Neuroevolution is a competitive alternative to reinforcement learning for skill discovery. ICLR, 2023.
>
> [2] QDax: A library for quality-diversity and population-based algorithms with hardware acceleration. arXiv, 2023.
>
> [3] [https://qdax.readthedocs.io/en/latest/examples/pgame/](https://qdax.readthedocs.io/en/latest/examples/pgame/)
>
> [4] Evolving populations of diverse RL agents with MAP-Elites. ICLR, 2023.
>
> [5] [https://qdax.readthedocs.io/en/latest/examples/me_sac_pbt/](https://qdax.readthedocs.io/en/latest/examples/me_sac_pbt/)
>
> [6] The quality-diversity transformer: Generating behavior-conditioned trajectories with decision transformers.
>
> **We hope that our response has addressed your concerns, but if we missed anything please let us know.**

---

> > ### Comment · Reviewer_CbDL · 2023-11-21
> > **Good rebuttal**
> >
> > I thank the authors for their impressive rebuttal work. The explanations provided alongside with the new experimental results added to the paper sound convincing to me. I am happy to change my recommendation to accept.

---

### Official Review · Reviewer_zwMj · 2023-10-31

**Soundness:** 4 excellent
**Presentation:** 3 good
**Contribution:** 3 good
**Rating:** 8
**Confidence:** 3

**Summary:**

**Problem Setting**

This work aims to tackle the methods of quality diversity, where we desire to find a diverse set of solutions to an optimization problem. Specifically, we try to make QD methods more computationally efficient.

**Novel Idea**

The idea is to factor the population space of all neural network parameters into two spaces, the first layers and the later layers. These correspond to a representation layer and a decision layer.

The representation part has a smaller population and is implemented as a list. The decision portion contains a much larger population and is impllemented as a grid.

Parent selection is performed uniformly. Parts are selected independently from the decision and representation portions, then evaluted jointly. The mutation operator is implemented through a combination of an evolutionary operator and direct optimization through a critic. There is a unique critic for each representation part. Survivors are selected through the vanilla MAP Elites style approach where parts are first segmented into cells, and then chosen based on fitness.

Experiments are presented on the QDax suite. Comparisons are presented against map elites, along with modern versions. Experiments show strong results and CCQD consistently outperforms the prior methodology.

**Strengths:**

This paper presents a simple method to reduce the sample complexity of quality diversity methods applied to neural networks. The idea is to partition the parameter space into a representation layer, which consists of the first neural network layers, and the decision layer, which is the later laters. This is a clean formulation and is communicated in an understandable manner. The significance of this work is in its further exploration as a method to factorize evolutionary methods over neural networks.

Ablations provide answers to questions such as what the qualitative impact of the representation vs the decision parts are.

**Weaknesses:**

The setting provided places an emphasis on sample complexity, without considering the asymptotic performance of the methods. It would be strengthening to confirm that the CCQD method reaches the same or better asymptotic performance as the other methods.

In addition, a description of the diversity criteria used to separate the symbols in the QD algorithm would improve clarity.

**Questions:**

What is the motivation between using both an evolutionary operator as well as a learned actor critic update?

It would be interesting to see if there is a structured way to decompose the neural network into factors to be used for QD. It would also be informative to see the affect of QD on the parameters themselves, i.e. are decision policies trained via QD more robust to noisy inputs because they are trained on a variety of representations?

---

> ### Author Response · Authors · 2023-11-20
> **Response by the authors**
>
> Thank you for your appreciation. Below please find our response.
>
> ## Q1 Asymptotic performance
>
> Thanks for your suggestion. We have revised to increase the timesteps of each environment by three times, i.e., from 5e7 to 1.5e8, to show the asymptotic performance. The results (in Table 1, Figure 2, and Table 8) indicate that CCQD remains the best-performing algorithm across various metrics, e.g., QD-Score AUC and Max Fitness.
>
> ## Q2 Description of the diversity criteria
>
> Thanks for your suggestion. We have revised the description of the QD algorithm in Appendix A according to your suggestion. Specifically, we add a detailed process of survivor selection of QD to make it clearer.
>
> ## Q3 Motivation of using both an evolutionary operator as well as learned actor critic update?
>
> The RL operator focuses on optimizing quality. To promote diversity optimization, it is necessary to simultaneously use evolutionary operators. This is a common approach in many QDRL works [1-3]. Thanks to your comment, we have revised to add some explanation. Furthermore, we analyzed the effectiveness of the evolutionary operator in Figure 10 of Appendix C.7. We can observe that as the optimization proceeds, the proportion of RL solutions (i.e., solutions generated by the RL operator) surviving into the archive in each generation tends to decrease, i.e., the proportion of surviving solutions generated by evolutionary operator tends to increase, which demonstrate its effectiveness.
>
> ## Q4 Structured way to decompose the NN.
>
> Thanks for your valuable comments. Our proposed layer-based decomposition strategy can be seen as an initial step towards this. As you suggested, there may be some structured ways to decompose the neural network. We will study it in our future work.
>
> ## Q5 Are the decision policies more robust to noise because they are trained on a variety of representations?
>
> Yes, we agree with you. Some studies in zero-shot coordination and robust training [4-6] have shown that training with diverse partners can lead to robust coordination abilities, which is similar to the situation described here. Therefore, we believe that the decision policies trained in this manner are more robust. We will explore this further in our future work. Thank you.
>
> [1] Policy gradient assisted MAP-Elites. GECCO, 2021.
>
> [2] Empirical analysis of PGA-MAP-Elites for neuroevolution in uncertain domains. ACM TELO, 2022.
>
> [3] Evolving populations of diverse RL agents with MAP-Elites. ICLR, 2023.
>
> [4] Trajectory diversity for zero-shot coordination. ICML, 2021.
>
> [5] Learning zero-shot cooperation with humans, assuming humans are biased. ICLR, 2023.
>
> [6] Robust multi-agent coordination via evolutionary generation of auxiliary adversarial attackers. AAAI, 2023.
>
> **We hope that our response has addressed your concerns, but if we missed anything please let us know.**

---

### Author Response · Authors · 2023-11-20
**General Response**

We are very grateful to the reviewers for carefully reviewing our paper and providing constructive comments and suggestions. We have revised the paper carefully according to the comments and suggestions, where the changed parts are colored in red. Our response to individual reviewers can be found in the personal replies, but we would also like to make a brief summary of revisions for your convenience.



1. Related work

   1. We add discussions about some missing related works.

   2. We have revised to move some technical details that are common to QDRL in Section 2.3 to the appendix and added more discussions about the co-evolution, according to the suggestion of Reviewer CbDL.

2. Experiments

   1. We run all methods longer, increasing the timesteps of each environment by three times, i.e., from 5e7 to 1.5e8, in order to compare the asymptotic performance of different algorithms. The results indicate that CCQD remains the best-performing algorithm across various metrics, e.g., QD-Score AUC and Max Fitness. Due to the tight time limitation, we mainly run the main experiments and will run the other experiments in the Appendix later.

   2. In our previous experiments, due to our episode length being 250, the scaling of the Max Fitness was significantly smaller compared to the setting with an episode length of 1000. We additionally conduct all the experiments with episode length of 1000. The results still confirm the effectiveness of our algorithm.

   3. We re-run PBT-ME using the setting as described in the original paper and update the corresponding results in our paper. The results of PBT-ME under the two settings are quite similar. Under our setting, the Max Fitness is slightly higher, while under the original setting, the QD-Score is slightly higher.





**We hope that our response has addressed your concerns, but if we missed anything please let us know.**

---

### Meta-Review · Area_Chair_3Qwa · 2023-12-13

**Metareview:**

Very interesting paper with clear novelty (using cooperative coevolution in quality-diversity) and good results. Combining cooperative coevolution and QD strikes me as a very good idea in its own right, and very fertile. The paper is worth accepting for that reason alone. Also, it is well written, and has appropriate and extensive discussion of its intellectual precursors.

My main concern is to what extent this particular decomposition of the neural network will generalize. But I guess that's for future work.

**Justification For Why Not Higher Score:**

Not clear how well the specific method will generalize.

**Justification For Why Not Lower Score:**

Very clever idea that others can build on; well-written and generally well-executed paper.

---

### Decision · Program_Chairs · 2024-01-16

Accept (spotlight)